# Agricultural impacts on streams near Nitrate Vulnerable Zones: A case study in the Ebro basin, Northern Spain

**Rubén Ladrera**[1], **Oscar Belmar**[2], **Rafael Tomás**[1], **Narcís Prat**[3], **Miguel Cañedo-Argüelles**[3] *

**1** Food and Agriculture Department, Science and Technology Complex, University of La Rioja, Logroño, La Rioja, Spain, **2** Marine and Continental Waters Program, IRTA, Sant Carles de la Ràpita, Catalonia, Spain, **3** Freshwater Ecology, Hydrology and Management Group (FEHM), Departament de Biologia Evolutiva, Ecologia i Ciències Ambientals, Institut de Recerca de l'Aigua (IdRA), University of Barcelona, Catalonia, Spain

* mcanedo.fem@gmail.com

**Data Availability Statement:** All relevant data are within the paper and its Supporting Information files.

## Abstract

Agricultural intensification during the last century has caused river degradation across Europe. From the wide range of stressors derived from agricultural activities that impact rivers, diffuse agricultural pollution has received most of the attention from managers and scientists. The aim of this study was to determine the main stressors exerted by intensive agriculture on streams around Nitrate Vulnerable Zones (NVZs), which are areas of land that drain into waters polluted by nitrates according to the European Nitrate Directive (91/676/EEC). The study area was located in the NW of La Rioja (Northern Spain), which has some of the highest nitrate concentrations within the Ebro basin. The relationships between 40 environmental variables and the taxonomic and functional characteristics of the macroinvertebrate assemblages (which are useful indicators of water quality) were analyzed in 11 stream reaches differentially affected by upstream agricultural activity. The streams affected by a greater percentage of agricultural land cover in the surrounding catchment had significantly higher nitrate concentrations than the remaining sites. However, hydromorphological alteration (i.e. channel simplification, riparian forest and habitat degradation), which is closely linked to agricultural practices, was the main factor affecting macroinvertebrate assemblages. We suggest that "good agricultural practices" should be implemented in streams affected by NVZs to reverse stream degradation, in concordance with the European Water Framework Directive (WFD).

## Introduction

Agriculture is one of the most important pressures affecting freshwater ecosystems around the world [1,2]. Intensive agriculture: i) strongly affects water quality through pesticides and fertilizers, mainly nitrates [3]; ii) requires large quantities of water for irrigation, and iii) degrades the fluvial habitat through riparian forest removal, channel incision and straightening, reduction of bank stability and sediment deposition [4].

**Funding:** This study received funding from the Institute of Studies of La Rioja, which did not play a role at any point in the research or publication process. Miguel Cañedo-Argüelles was supported by the MECODISPER project (CTM2017-89295-P) funded by the Spanish Ministerio de Economía, Industria y Competitividad - Agencia Estatal de Investigación and cofunded by the European Regional Development Fund (ERDF). The funders had no role in study design, data collection and analysis, decision to publish, or preparation of the manuscript.

**Competing interests:** The authors have declared that no competing interests exist.

To revert agricultural impacts on streams, different Directives [i.e. a legal act requiring member states to achieve a certain result) have been implemented in the European Union (EU) during the last decades. One example is the Nitrates Directive (91/676/EEC; ND), which was adopted in 1991. The ND aims to protect water quality across Europe reducing nitrates from agricultural sources and promoting the use of good farming practices. Within the ND, Nitrate Vulnerable Zones (NVZs) are defined as the areas of land that drain into waters polluted by nitrates, and farmers with land in these zones have to follow mandatory rules to restore water quality [5]. The ND forms an integral part of the European Water Framework Directive (2000/60/EC; WFD), which is aimed at establishing a framework for water protection, so all the water bodies in Europe reach a "good ecological status" by 2021 or 2027. To achieve this good ecological status, the WFD indicates that pressures and impacts at the basin scale must be identified to guide management measures. This identification is often inadequate in existing monitoring programs [6].

The European Water Framework Directive (WFD) also establishes the need to use biotic indices to assess the ecological status of rivers, and the most widely used indices are those based on macroinvertebrate assemblages [7–9]. Regarding agriculture, there is an extensive literature related to assessing its impacts on rivers through macroinvertebrate assemblages, using both the taxonomic composition and the functional structure [4,10–15]. Compared with the taxonomic approach, the study of the functional structure (i.e. trait-based approaches) presents a higher temporal and spatial stability across regions because the variability attributed to species composition of different areas is reduced [16,17].

Various studies suggest that 10 ppm of nitrate can adversely affect, at least during long-term exposures, freshwater invertebrates [18,19]. However, beyond nitrate toxicity, nitrate pollution in rivers and streams has been associated with other environmental factors, for example increased autotrophic production due to reduced shading of the stream channel [20,21]. High periphyton biomass reduces oxygen concentrations in the water-column and the substrate interstices [22,23], with increased ecosystem respiration leading to impacts on pollution sensitive species such as some Ephemeroptera, Plecoptera and Trichoptera (EPT) [24]. Excessive algal growth can also smother the riverbed, reducing substrate availability for grazing mayflies, which are replaced by molluscs, oligochaetes and chironomids [22].

Additionally, stream macroinvertebrate assemblages located in agricultural areas are especially sensitive to habitat deterioration, which is mainly related with an increase in fine sediments [25–27]. Fine sediments remove and homogenize habitats [27], leading to reduced taxa richness, especially some EPT species [26,27], and altered assemblage composition [28]. Sedimentation also alters the functional structure of macroinvertebrate assemblages [13,15,25,29–31]. Higher amounts of sediment can cover vegetal and mineral coarse substrate leading to an increase in burrowers, and a decrease in crawlers and scrapers [13,15,30–33].

The present study assesses the potential impact of agriculture on stream ecosystems around NVZs using the taxonomic and functional composition of macroinvertebrate assemblages as indicators. The Ebro basin was selected as a case study because of being subjected to heavy human pressures, especially agriculture [34,35]. The specific objectives of the study were to: i) determine the main agricultural stressors on streams located in areas affected by intensive agriculture around NVZs; and ii) assess their degree of compliance with the WFD according to physicochemical variables and macroinvertebrate indices. We predicted that these streams would be affected not only by nitrate pollution, but also by an increase in fine sediments and a reduction in riparian and aquatic habitat quality. As a consequence, we expected that agricultural intensification would lead to a shift in the taxonomic composition and functional structure of aquatic macroinvertebrate assemblages, from assemblages dominated by pollution-

sensitive species that feed on coarse organic matter (e.g. leaf litter) to assemblages dominated by tolerant species feeding on fine organic matter (e.g. suspended particles) and algae.

## Methodology

### Study area and sampling strategy

The study area is located in the NW of La Rioja (northern Spain, Ebro Basin; Fig 1), in the Oja-Tirón, Zamaca and Najerilla sub-basins. To reduce natural differences among sites, the study was carried out in six first-order streams (Fig 1) belonging to the typology "Calcareous Mediterranean mountain rivers" according to the classification developed in Spain to fulfil the WFD (ORDER ARM/2656/2008, Ministry for the Environment). Eleven sites (100 m length) were studied. All of them were free from significant urban or industrial effluents in the adjacent area and they mainly differed in the upstream agrarian activity (i.e. percentage of cultivated land). There are no large villages (i.e. more than 500 inhabitants) in the area and almost no industrial activity. The study sites were selected to fulfill the above-mentioned criteria after a visual inspection of the region. The information on the location of the sampling sites is provided in the supplementary material (Table A in S1 File).

Well-preserved forests exist in the upper course of Yalde and Reláchigo Streams (Fig 1), where agricultural activity is limited. The cultivated area increases as the streams flow into the Ebro Valley, where intensive farming dominates (mainly cereal, vineyard and horticultural crops). As a result of the intense agricultural activity, two NVZs exist in the study area, called "Zamaca and Oja alluvial" and "Low Najerilla alluvial". The six streams selected in the present study are the only first-order around these NVZs. According to monitoring programs carried

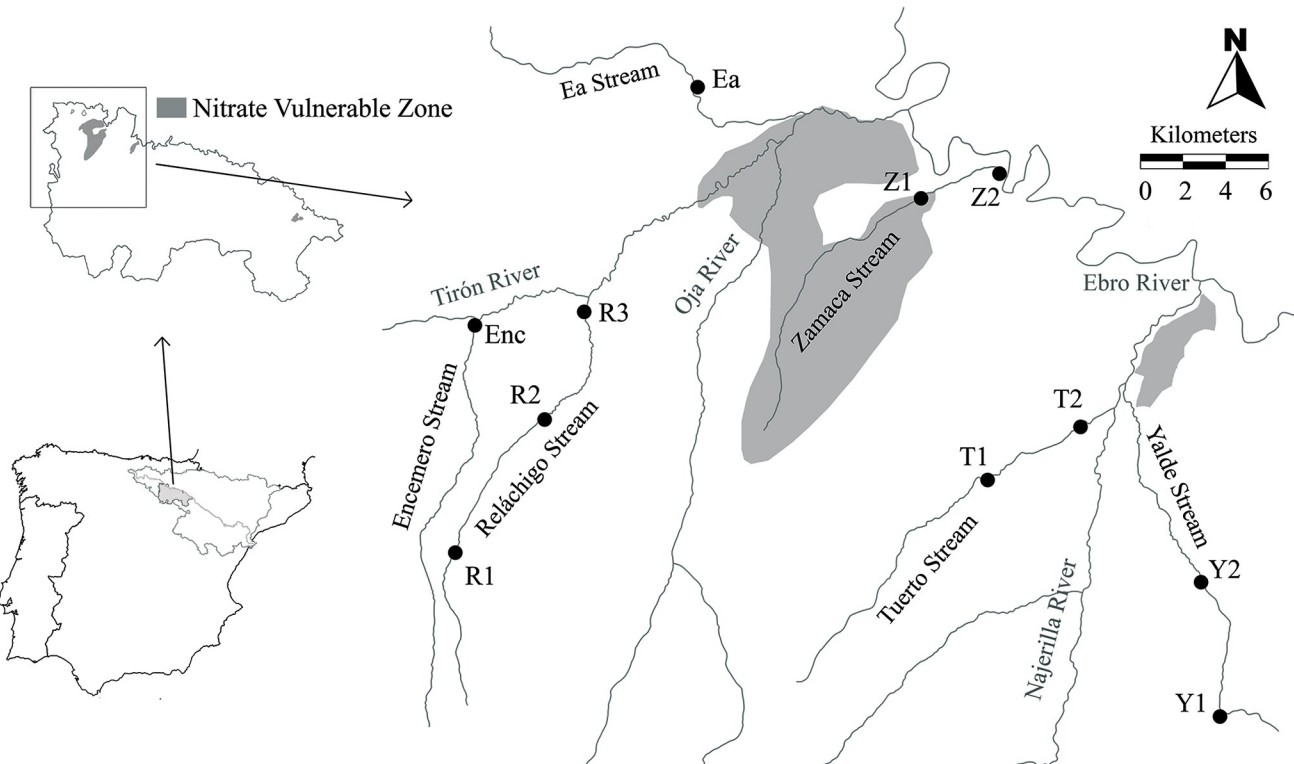

**Fig 1. Map of the study area.** Sampling sites in the study area (La Rioja, Spain). Nitrate Vulnerable Zones (NVZs) are coloured in grey ("Zamaca and Oja alluvial" and "Low Najerilla alluvial").

out by the Ebro Hydrographic Confederation (CHE; http://www.chebro.es/), nitrate concentrations in the study area had exceeded the highest value established by WFD to consider a river or stream under a good ecological status. However, some of these streams have been studied just sporadically and all of them just at one site.

Each site was sampled four times during 2017 (i.e., February, May, August, and November). Physicochemical variables were analyzed at each sampling occasion, whereas habitat quality and macroinvertebrates were sampled only in May.

## Environmental variables

Forty environmental variables belonging to different categories (physicochemical, hydromorphological, geological, land cover and topographic; Table B in S1 File) were studied for each site.

Water temperature (˚C), pH, dissolved oxygen (ppm) and electrical conductivity (mS/cm) were measured *in situ* using handheld probes (PCE-PHD1 multifunctional instrument). For ion analysis, 125 ml water samples were collected and transported in a cool-box to the laboratory, where they were frozen until being analyzed using chromatographic standard methods [36].

The riparian habitat of each site (100 m length) was inspected and characterized using the four components of the QBR Riparian Forest Quality Index by Munné *et al*. [37]: total riparian vegetation cover, cover structure, cover quality and channel alterations (Table B in S1 File). The fluvial instream habitat was also inspected and characterized using the IHF River Habitat Index [38], which measures habitat heterogeneity by analyzing seven features: hard substrate embeddedness in fine sediment, riffle frequency, substrate composition, velocity/depth conditions, shading of river bed, heterogeneity components (leaf litter, wood, roots or natural dams), and aquatic vegetation cover (Table B in S1 File). Both indexes range from 0 to 100 with high scores associated with better habitat conservation. Additionally, the percentage of macrophyte cover was visually determined.

The geological, land cover and topographic variables were respectively retrieved from the Spanish Geological Survey (IGME), the Spanish Land Cover and Use Information System (SIOSE) and the Spanish National Geographic Institute (IGN) using a Geographic Information System (GIS).

## Macroinvertebrate sampling and biotic indices

Macroinvertebrates were collected according to the Standard protocol for benthic invertebrates of wadeable streams in Spain (ML-Rv-I-2013; [37]). This quantitative protocol considers the presence of five different habitat types: hard substrate, fine sediments, plant debris, submerged bank vegetation and submerged macrophytes. Twenty Surber samples (0.1 m$^2$) were taken at each site, proportionally distributed among each type of habitat according to the surface that it occupied within the streambed. The permission to sample was issued by the Government of La Rioja. The collected material was stored in one sample and preserved in 70% ethanol and taken to the laboratory to be identified. The identification of macroinvertebrates was generally done to genus level, except for some Diptera families (subfamily level) and Oligochaeta. Sub-sampling was conducted to estimate the taxa abundances when 300 individuals per sample were counted, although sample exploration continued to check for new taxa.

The IBMWP (Iberian Biological Monitoring Working Party; [39]) and IMMi-T (Iberian Mediterranean Multimetric Index, using quantitative data [40]) biotic indices were calculated using the MAQBIR software [41]. The IBMWP is a qualitative index based on the tolerance of macroinvertebrate families to water quality and river alteration. The presence of each family

provides a single score out of 10 (being 1 highly tolerant and 10 highly sensitive) with the cumulative scores providing the final IBMWP score. The IMMi-T is a quantitative multimetric index based on the number of families, EPT, IASPT (Iberian Average Score per Taxon, [42]) and log (selected EPTCD + 1), where EPTCD is the number of sensitive Ephemeroptera, Plecoptera, Trichoptera, Coleoptera and Diptera families. Following the WFD, this index was standardized by calculating the EQR (Ecological Quality Ratio) value dividing the value of each metric by the reference value for Calcareous Mediterranean mountain rivers.

## Functional traits

Macroinvertebrate functional structure were characterized using five biological and two ecological traits, containing 35 and 12 categories respectively (Table C in S1 File; [43,44]). Biological traits describe life-cycle features ("reproductive cycles per year" trait), physiology ("respiration"), resilience or resistance potentials ("locomotion and substrate relation") and feeding behavior ("food, and feeding habits"), which are the four groups of biological features described by Usseglio-Polatera *et al.* [43]. The ecological traits reflect the organism's environmental preferences and the behaviours associated with these preferences. Here we used two ecological traits referring to the microscale characteristics of the habitat used by invertebrates ("microhabitat") and to the trophic status of the freshwater that they inhabit ("trophic status"). The selected traits have been extensively used in previous studies related to agricultural and other land use impacts on rivers (e.g. [15,27,29,30,44–46]), as well as fine sediment increase [31]. These ecological traits were useful to assess nutrient and hydromorphological impacts on the streams, whereas the traits "locomotion and substrate relation", "food" and "feeding habits" have been previously used as effect traits (i.e. those that have a direct influence on a specific function of the ecosystem [47–49]).

With this information, a dataset of relative abundance of trait categories per sample was built following a 'fuzzy coding' approach [50]. The trait categories [44] were assigned to an affinity score for each taxa ranging from 0 to 5, from null to high affinity, respectively [50].

## Statistical analysis

A hierarchical cluster analysis of the macroinvertebrate assemblages (unweighted pair group method using arithmetic averages, UPGMA; PRIMER, [51]) was done to group sites by their similarity. Then, environmental variables were examined within these groups.

An Indicator Species Analysis (IndVal, [52]) was used to determine the invertebrate taxa significantly associated with each of the groups previously established. This method provides an indicator value (IV-value) for each taxon according to its presence and abundance in each group. A Monte Carlo permutation test with 9999 permutations was used to test the significance of each IV-value (P<0.05). For each trait category, a nonparametric pairwise Mann-Whitney U test was performed to detect differences between the groups defined by the cluster analysis. Asymmetric beanplots (software R3.4.2, package beanplot, [53]) were used to visualize the differences in trait abundance between groups.

In order to determine the main environmental variables related to differences in macroinvertebrate assemblages among sites, a DISTLM (distance-based linear model) analysis was performed (PERMANOVA + for PRIMER, [54]) based on genera abundances. DISTLM is a routine for analysing and modelling the relationship between a multivariate data cloud, as described by a resemblance matrix, and one or more predictor variables [54]. In our case, the resemblance matrix described dissimilarities among macroinvertebrate samples of studied sites, based on genus abundance data, and the predictor variables were the environmental variables (Table B in S1 File).

The macroinvertebrate distance matrix was created using the Bray Curtis distance after the data were log $(x + 1)$ transformed. The environmental variables were log $(x + 1)$ transformed and normalized. Variables that were highly correlated with others (Spearman correlation coefficient higher than 0.9) were removed from the analysis. The values of the physicochemical variables included in the analysis were the averages of the four sampling dates since they represent better the physicochemical state of the streams and their influence on the biological assemblages.

DISTLM provides quantitative measures and tests of the variation explained by one or more predictors, allowing the partitioning of variance according to a regression (or multiple regression) model [54]. The DISTLM routine was performed using a step-wise selection procedure based on the AIC selection criteria [55], to obtain a set of variables plausibly explaining changes in the composition of the macroinvertebrate assemblages. From every variable included in the final model just some of them significantly explain the variation in the aquatic macroinvertebrate assemblages between sites differentially affected by agriculture. The final model from DISTLM analysis was visualized by a distance-based Redundancy Analysis (dbRDA) plot. The dbRDA routine performs a constrained ordination of the macroinvertebrate assemblages data using the DISTLM final model.

## Results

### Biotic, environmental and hydromorphological characteristics

Two groups of sampling sites were differentiated based on the hierarchical cluster analysis of the macroinvertebrate assemblages (Fig 2). The two groups presented a 60% of dissimilarity

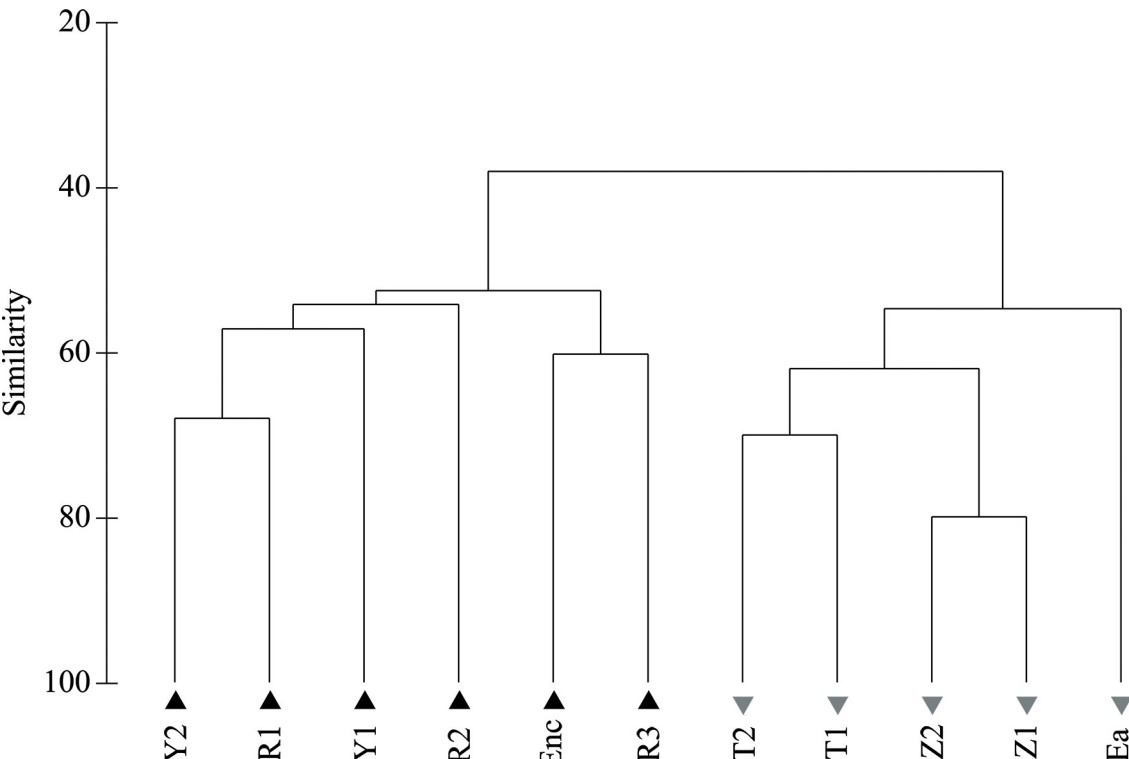

**Fig 2. Hierarchical cluster.** Hierarchical cluster analysis (UPGMA) based on the composition of the macroinvertebrate community of each site. The resultant groups are identified with black triangles (group 1) and grey inverted triangles (group 2).

and separated sites with low (<50%) and high (>70%) agricultural -or cultivated- land cover upstream the river reach (Table 1).

Regarding the environmental variables studied and included in the WFD (Table 1), nitrate showed particularly high concentrations at different dates and sites. Considering average values, 4 sites exceeded 25 ppm, the highest value established by RD 817/2015—Spanish legislation based on WFD—to consider a river or stream in a good ecological status (R3, Ea, Z1 and Z2; Table 1). However, there was a great variability in nitrate concentrations among dates, as a concentration of 25 ppm was exceeded at least once in every site of group 2 (Ea, Z1, Z2, T1 and T2) during the study (Table 1). The lowest values of the IBMWP and IMMi-T biotic indices were recorded at group 2, and they were below the good ecological status threshold for Z1, T1 and T2 (Table 1).

Sites of group 2 also showed the lowest values of the IHF and QBR indices (Table E in S1 File). The first component of the IHF, referred to the embeddedness of hard substrate in fine sediment (Table E in S1 File), had a score of 0 at every site of group 2, since more than 60% of boulders, cobbles and pebbles were embedded in fine sediment. Macrophytes occupied between 30 and 60% of the streambed at all study sites, except for R2 (20%) and for Y1, Y2 and R1, where macrophyte cover was less than 5% (Table E in S1 File).

Among the other environmental variables studied, the average conductivity resulted particularly high at Enc site (mean conductivity 2077 μS/cm). Accordingly, exceptionally high values of calcium and sulfate were recorded at this reach (Enc; sulfate = 896.79 ppm; calcium = 575.52 ppm; Table C in S1 File).

## Macroinvertebrate assemblage and agriculture stressors

Group 1 was mainly characterized by EPT taxa such as Polycentropodidae (*Plectrocnemia sp.*), Heptageniidae (*Ecdyonurus sp. and Rithrogena sp.*), Leptophlebiidae and Leuctridae (*Leuctra sp.*), which are sensitive to pollution according to the IBMWP index (Table 2). Other taxa associated with this group were Gammaridae, Ceratopogonidae and the beetles *Esolus* and *Elmis* (Elmidae). Group 2 was significantly associated with taxa with a low IBMWP score (Table 2),

**Table 1. Environmental variables.** Values of the physicochemical variables and the biotic indices -IBMWP (Iberian Biological Monitoring Working Party) and IMMi-T (Iberian Mediterranean Multimetric Index)- used to assess the good ecological status of streams according to the Spanish legislation (RD 817/2015), based on the European Water Framework Directive (WFD). The limits established by the Spanish legislation to consider a water body at good ecological status or higher for "Calcareous Mediterranean mountain rivers" are shown in parentheses. For chemical variables, mean values and ranges (i.e. minimum and maximum) are shown. Values exceeding the threshold for the good ecological status according to the RD 817/2015 are highlighted in bold. Asterisk indicate that although the mean values were below the threshold, the threshold was exceeded at least once during the study period. The lowest detectable values for phosphate and ammonia were 0.010 and 0.020, respectively. IHF (River Habitat Index) and QBR (Riparian Forest Quality Index) values, which are not considered by RD 817/2015 to determine the good ecological status of a water body, are shown at the right part along with the percentage of area occupied by agricultural land upstream the study site (AGR). The ranges for IHF and QBR are shown in parentheses, with higher scores associated to better habitat conservation. The lowest values of IHF (<50) and QBR (<50) and the highest values of AGR (≥70) are highlighted in bold.

| | | (6–9) | (5 ppm) | (25 ppm) | (0.4 ppm) | (0.6 ppm) | -93 | -0.69 | (0–100) | (0–100) | (%) |
|---|---|---|---|---|---|---|---|---|---|---|---|
| Group 1 | Enc | 8.07 (7.95–8.21) | 9.20 (7.30–10.20) | 20.14 (7.01–38.86)* | 0.010 (0.010–0.010) | 0.055 (0.020–0.119) | 134 | 0.91 | 67 | 50 | 46.6 |
| | R1 | 7.92 (7.76–8.01) | 10.17 (9.50–10.70) | 0.95 (0.57–1.86) | 0.014 (0.010–0.017) | 0.187 (0.150–0.205) | 207 | 1.15 | 74 | 75 | 1 |
| | R2 | 8.16 (8.04–8.15) | 8.80 (7.80–9.80) | 2.58 (1.46–4.08) | 0.010 (0.010–0.011) | 0.171 (0.143–0.192) | 129 | 0.93 | 71 | 50 | 17.6 |
| | R3 | 8.38 (8.18–8.50) | 9.67 (8.30–10.40) | **41.85 (15.20–108.65)** | 0.036 (0.010–0.066) | 0.069 (0.020–0.126) | 147 | 0.94 | 60 | **40** | 48.1 |
| | Y1 | 8.41 (8.27–8.54) | 9.50 (9.10–10.30) | 0.63 (0.43–0.96) | 0.009 (0.010–0.010) | 0.165 (0.145–0.191) | 220 | 1.3 | 71 | 80 | 0 |
| | Y2 | 7.52 (7.13–8.30) | 6.90 (5.70–7.90) | 4.12 (2.47–6.15) | 0.013 (0.010–0.018) | 0.159 (0.020–0.215) | 162 | 1.04 | 57 | 75 | 7.4 |
| Group 2 | Ea | 8.01 (7.96–8.05) | 7.77 (7.40–8.30) | **30.55 (14.12–74.80)** | 0.148 (0.010–0.438)* | 0.050 (0.020–0.128) | 112 | 0.76 | **42** | **40** | **79.2** |
| | Z1 | 8.11 (8.00–8.21) | 9.17 (8.70–9.80) | **88.83 (79.42–94.57)** | 0.044 (0.010–0.150) | 0.135 (0.020–0.194) | **92** | **0.67** | **32** | **10** | **84** |
| | Z2 | 8.22 (8.13–8.35) | 9.00 (8.80–9.30) | **84.68 (73.54–103.74)** | 0.074 (0.010–0.175) | 0.019 (0.020–0.020) | 108 | 0.73 | **36** | **10** | **84.2** |
| | T1 | 7.85 (7.62–8.10) | 6.93 (4.30–9.50)* | **21.13 ± (5.01–57.91)*** | 0.136 (0.010–0.246) | 0.156 (0.072–0.200) | **59** | **0.35** | **29** | **10** | **70** |
| | T2 | 8.16 (8.00–8.35) | 8.13 (6.10–10.30) | 19.77 ± (8.08–38.34)* | 0.312 (0.010–0.684)* | **1.584 (0.020–5.016)** | **56** | **0.33** | **31** | 35 | **76.8** |

**Table 2. IndVal analysis.** IndVal analysis results showing taxa significantly associated with one of the previously established groups (Fig 2). For those taxa included in the IBMWP index, their score is shown. # means that the whole family was significantly associated with one of the groups, but no genus within the family was.

| Taxa | Group | IV value | P | Freq | IBMWP value |
|---|---|---|---|---|---|
| Heptageniidae | 1 | 1 | 0.004 | 6 | 10 |
| *Ecdyonurus* | 1 | 0.833 | 0.023 | 5 | |
| *Rithrogena* | 1 | 0.667 | 0.05 | 4 | |
| Polycentropodidae | 1 | 1 | 0.001 | 6 | 10 |
| *Plectrocnemia* | 1 | 0.833 | 0.016 | 5 | |
| Leptophlebiidae # | 1 | 0.833 | 0.023 | 5 | 10 |
| Leuctridae | 1 | 0.833 | 0.016 | 5 | 10 |
| *Leuctra* | 1 | 0.833 | 0.023 | 5 | |
| Gammaridae | 1 | 0.802 | 0.005 | 9 | 6 |
| Ceratopogonidae | 1 | 0.793 | 0.024 | 6 | 4 |
| Ceratopogoninae | 1 | 0.793 | 0.035 | 6 | |
| Elmidae | 1 | 0.69 | 0.012 | 9 | 5 |
| *Esolus* | 1 | 0.957 | 0.001 | 7 | |
| *Elmis* | 1 | 0.81 | 0.018 | 7 | |
| Hydroptilidae | 2 | 0.8 | 0.01 | 4 | 6 |
| *Hydroptila* | 2 | 0.8 | 0.012 | 4 | |
| Haliplidae | 2 | 0.788 | 0.042 | 8 | 4 |
| *Haliplus* | 2 | 0.788 | 0.039 | 8 | |
| Psychodidae | 2 | 0.755 | 0.047 | 9 | 4 |
| Simuliidae | 2 | 0.754 | 0.03 | 8 | 5 |
| Simuliini | 2 | 0.754 | 0.027 | 8 | |
| Baetidae | 2 | 0.631 | 0.013 | 10 | 4 |
| *Baetis* | 2 | 0.631 | 0.014 | 10 | |
| Oligochaeta | 2 | 0.622 | 0.038 | 11 | 1 |
| Chironomidae | 2 | 0.572 | 0.005 | 11 | 2 |
| Chironominae | 2 | 0.634 | 0.004 | 11 | |
| Orthocladiinae | 2 | 0.574 | 0.002 | 11 | |

like Oligochaeta and Chironomidae, together with the dipterans Simuliidae and Psychodidae and other taxa such as Hydroptilidae or Haliplidae.

According to the Mann Whitney test, functional trait categories significantly differed between groups 1 and 2 (Fig 3). Group 1 was characterized by mono- or semivoltine taxa, gill and plastron respiration, crawlers living on large substrate, shredder and scrapers feeding on microphytes, and taxa adapted to oligotrophic environments. Group 2 was characterized by polyvoltine taxa, living and feeding on fine sediment and macrophytes and adapted to eutrophic conditions (Fig 3).

The final model of the DISTLM explained 99.2% of the total variance and it included the following significant (p<0.05) explanatory variables (ranked in order of importance): total value of the River Habitat Index (IHF; 43.9% of total variance explained), percentage of area occupied by agricultural land upstream the river reach (11.8%), nitrate concentration (9.4%), the ratio of valley width to channel width (8.6%) and the component of the riparian habitat index (QBR) referring to channel alterations (7.3%). Other variables included in the final model were not significant (p>0.05): the component of the QBR index referring to riparian vegetation cover (7.2%), percentage of land occupied by pasture upstream (4.3%), chloride concentration (4.2%) and macrophyte cover (2.2%) (Fig 4).

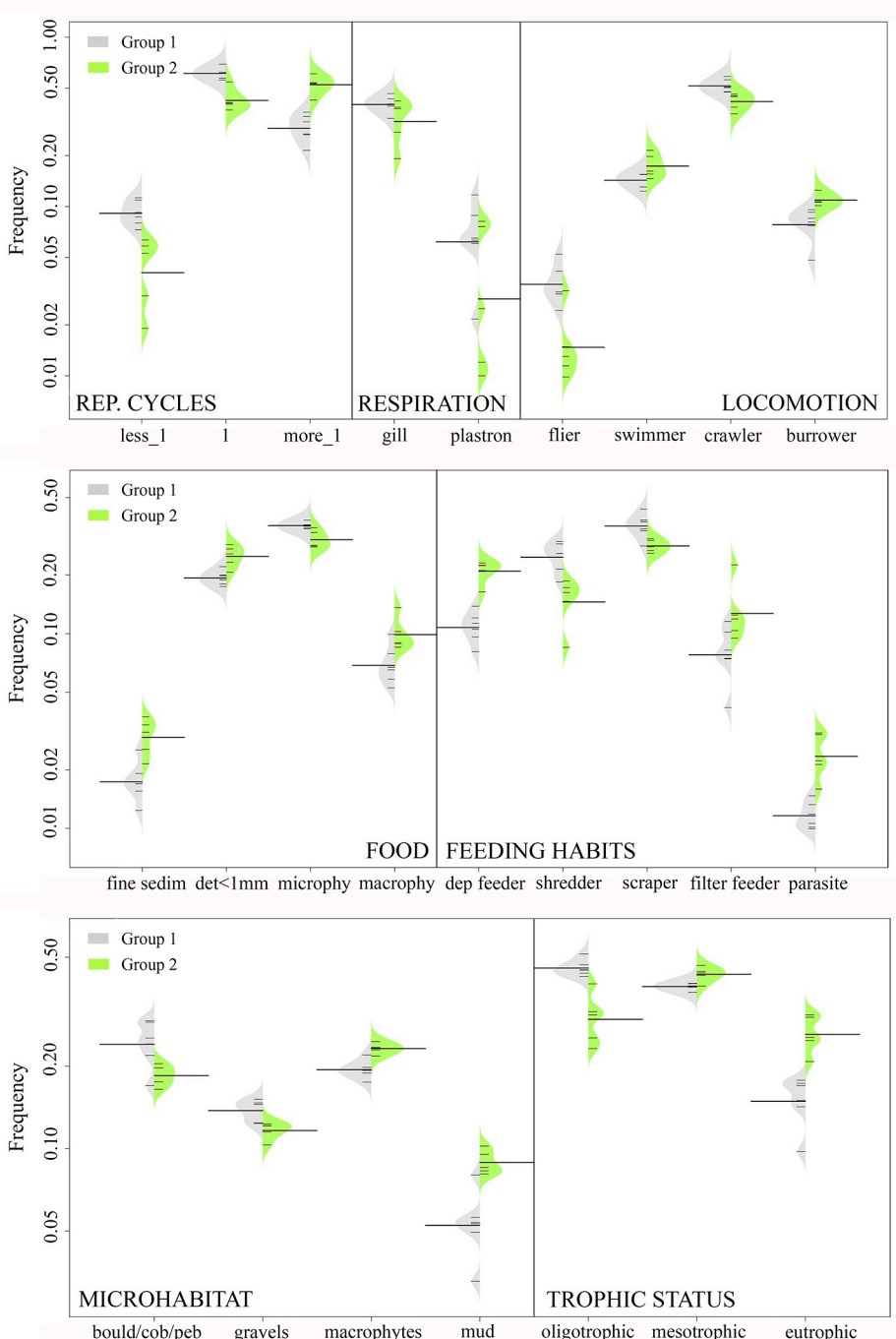

**Fig 3. Functional traits analysis.** Asymmetric beanplots of the functional trait categories significantly different (Mann Whitney U test, P<0.05) between the previously established groups (Fig 2). The individual observations of each site are shown as small white lines and the average of each group is shown as a black line.

## Discussion

Our results suggest that different stressors associated with agricultural activities could have a significant impact on the macroinvertebrate assemblages in the study region. As we hypothesized, and in agreement with previous studies [11,13,15,26], agricultural intensification led to a

**Fig 4. DISTLM analysis.** DISTLM analysis showing the dissimilarities between the macroinvertebrate assemblages of the studied sites, and predictor variables included in the final model -significant variables ($p<0.05$) are highlighted in bold-. Sites previously grouped by cluster analysis are shown with different symbols (black triangles: group 1; grey inverted triangles: group 2). IHF: Total score of the River Habitat Index; QBR-natur: component of the QBR referred to river channel alteration; $NO_3^-$: nitrate concentration; Cl: chloride concentration AGR: area occupied by agricultural land upstream the study site; VWI: Valley Width Index or ratio of valley width to channel width; PAS: area occupied by pasture upstream the study site.

replacement of EPT taxa by tolerant taxa such as Oligochaeta or Chironomidae. Likewise, and according to our hypothesis, streams were affected not only by nitrate pollution, but also by hydromorphological stressors like the alteration of riparian and aquatic habitat, leading to functional and taxonomic changes in the macroinvertebrate assemblages. Overall, sites with a higher upstream area occupied by agricultural land were dominated by polyvoltine taxa, a biological trait that has been associated with resistance strategies in human impacted sites [29,56].

Concentrations of nitrate significantly contributed to alter macroinvertebrate assemblages in the studied streams, as it has been observed in other agricultural areas [13,21]. Nitrate is able to convert oxygen-carrying pigments of aquatic animals to forms that are incapable of carrying oxygen (e.g. methemoglobin; [18,57]). In this regard, the mean nitrate concentration registered at seven of our study sites could be enough to adversely affect, at least during long-term exposures, freshwater invertebrates [18, 21]. However, nitrate uptake in aquatic animals is limited [58] and nitrate impact in aquatic ecosystems can be related to other processes, like an increase in autotrophic production [4]. Accordingly, macrophyte cover was higher in the

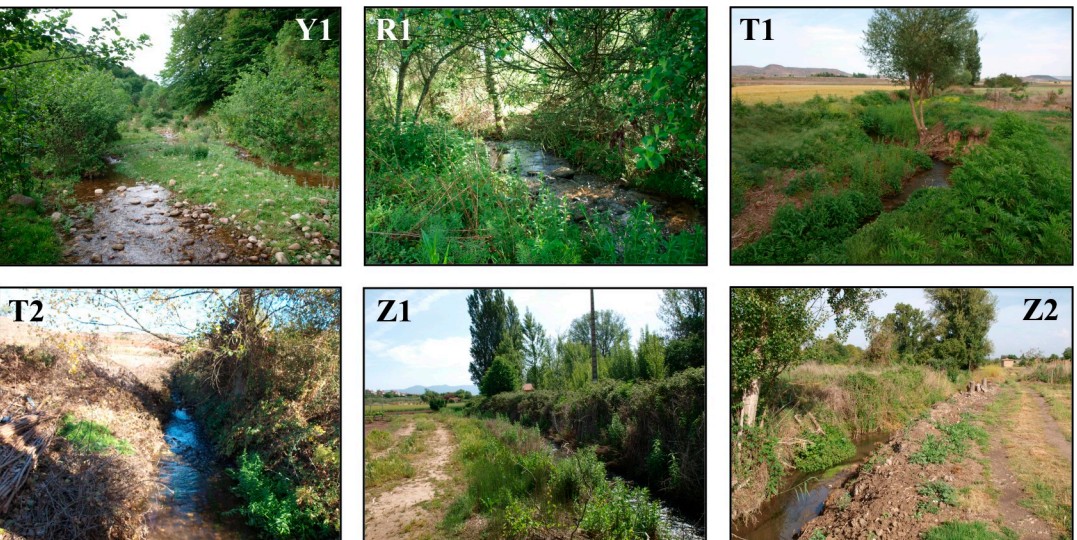

**Fig 5.** Photographs of the studied sites with the highest (white acronyms) and the lowest (black acronyms) QBR and IHF total values. Y1 (QBR = 80; IHF = 71), R1 (75; 71), T1 (10; 29), T2 (35; 31) Z1 (10; 32), Z2 (10; 36).

most agricultural impacted sites. This could be associated not only with a higher nitrate concentration, but also with the degradation of the riparian forest allowing more light to penetrate the streambed [59]. Overall, in these highly impacted streams, taxa living and feeding on macrophytes and adapted to eutrophic conditions were favored.

The variation in aquatic macroinvertebrate assemblages between sites was best explained by variables associated with the hydromorphological alteration of the stream habitat and with the riparian habitat. Specifically, the total IHF index, the QBR components that refer to riparian vegetation cover and river channel alteration and the ratio of valley width to channel width (VWI) were responsible for 67% of the variance in the macroinvertebrate assemblages. Different authors have pointed out that physical habitat degradation can have a greater influence on aquatic macroinvertebrates than the exposure to chemical pollution. For instance, Genito *et al.* [10] suggested that a high percentage of agricultural land cover caused a macroinvertebrate assemblage composition that reflected stream habitat degradation, whereas nitrate concentration had a weaker association with assemblage alteration. Shields *et al.* [60] pointed out the importance of stream habitat conservation in agricultural areas, especially related to channel degradation, large wood removal or increased deposition of fine sediments. Accordingly, fine sediment has been determined as a more pervasive stressor to macroinvertebrate assemblage than increased nutrient concentrations in streams around agricultural areas [27] and under controlled experimental conditions [15]. Beyond agricultural activities, several studies have confirmed the strong impacts of fine sediments on stream invertebrates (e.g. [61–64]).

Our results show that intensive agriculture can alter the streambed, simplify the stream habitats (i.e. high substrate embeddedness in fine sediment, low heterogeneity of substrates) and deteriorate the riparian forest (Fig 5). This can have important impacts on the aquatic ecosystem. For example, according to Genito *et al.* [10] and Piscart *et al.* [12], the reduction of riparian forests decreased the abundance of shredders in sites affected by intensified agricultural activity, leading to a lower litter breakdown rate. Riparian alteration also enhances, together with the intensive soil tillage, the amount of sediment entering the streams [4]. Although we did not measure sediment fluxes directly, the importance of the IHF index (both the overall score and the 1$^{st}$ component, referred to fine sediment accumulation) suggests that

sedimentation was an important explanatory variable of the functional structure of the macro-invertebrate assemblages in the study sites. Increased stream sedimentation has often been associated with nonpoint sources arising from agricultural land uses and it is mainly related to the loss of riparian vegetation [27].

In the present study, the greatest boulders, cobbles and pebbles embeddedness in fine sediment (i.e. component 1 of IHF index) were found at sites affected by a greater agricultural activity (Z1, Z2, T1, T2 and Ea). At these sites, there were less macroinvertebrates adapted to move on large substrate (i.e. crawlers), maybe because fine sediment decreased habitat heterogeneity and availability [27]. On the contrary, there were more invertebrates living in the mud and burrowers, as it has been previously reported for rivers experiencing sediment increases [15,30]. Agricultural intensification also favored organisms that fed on fine sediment, negatively affecting scrapers, as it was found by previous other studies [32,33]. This could be related to the sediment covering large substrates, thereby reducing microphytes' availability for stream fauna [65]. The abundance of shredders was also reduced under agricultural intensification. As suggested by Burdon et al. [27], this is most likely due to degraded riparian habitats (Fig 5), since shredders do not seem to be directly affected by sediment deposition [33]. Finally, respiration by plastron and gills were significantly less frequent in sites surrounded by a greater agricultural area. For example, we observed a decrease in *Elmis* and *Esolus*, which have larvae with anal gills and aquatic adults with plastron respiration [44,45]. Concordantly, species of Coleoptera dependent on a bubble or plastron to breath have shown to be sensitive to high sediment concentration in water [46].

Several of the studied sites did not meet the legal requirements to be considered in good ecological status according to the WFD. Three sites did not reach the good ecological status based on the IBMWP and IMMi-T biotic indices. Nitrate exceeded the concentration for a good ecological status at seven sites, despite all of them (except for Z1) were outside of NVZs. Accordingly, the official reports point to nitrate as the main agricultural stressor in the studied streams, although they only suggest the implementation of "good agricultural practices" at a few sites (and these practices are not detailed [66]). However, the WFD indicators seem to underestimate hydromorphological alterations, since they were not specifically considered when determining the ecological status of the water bodies. According to Houlden [67], hydromorphology is a "supporting element" in the WFD, which is only used to confer the high ecological status (i.e. a site cannot have the highest ecological status if it is hydromorphologically altered). As a consequence, despite the potential effect of hydromorphological alteration on aquatic macroinvertebrates, it has been largely overlooked by water managers within the Ebro basin [66].

Finally, it should be noted that the increased salinization, associated with fertilizers and inefficient irrigation in agricultural areas [68], should also be considered as a potential source of stress for stream organisms. However, in the study area, conductivity was below 1400 μS/cm at every site, except at Encemero stream (2078 ± 158 μS/cm). The high conductivity in Encemero is naturally associated with sodium sulfate tertiary deposits and secondary minerals, mainly gypsum and calcite [69]. Thus, our results suggest that secondary salinization is not severe in the study area and it probably had a weak effect on aquatic macroinvertebrates given the conductivity values recorded in this study [70].

## Conclusions and recommendations

Aquatic macroinvertebrates seem to be useful indicators of agricultural impacts, since both the taxonomic composition and mainly the functional structure responded to agricultural intensification. Intensive agriculture around NVZs in the study area significantly altered the stream

habitats and its associated macroinvertebrate assemblages. High nitrate content could be partially responsible for such alteration, but hydromorphological impacts, especially the increase in the amount of sediment that enters the streams, seem to be the main driver behind stream degradation. The macroinvertebrate assemblage's alteration caused by intensive agriculture includes significant changes in effects traits (locomotion, food and feeding habits), which can directly affect certain ecosystem services and properties as ecosystem engineering, nutrient cycling and resource processing [71,72].

The implementation of the ND and the WFD do not seem to be enough to prevent the degradation of streams in agricultural areas. As other authors did [5,73], we suggest to extend the NVZs in the study area since nitrate exceeded the limits established by the WFD to achieve the good ecological status in seven of the studied sites (despite 6 of them are located upstream of NVZs). Moreover, farming practices established by the regional administration in NVZs have not been effective to reduce nitrate pollution in these areas [74]. Thus, the compliance of these practices should be more rigorously controlled. Accordingly, Worral *et al.* [75] showed little improvements in nitrate content in surface water concentration of NVZs after 12–15 years in UK, although several limitations to assess the effectiveness of NVZs have been acknowledged [76,77]. It is necessary to improve water quality monitoring programs to identify and quantify the main pollution sources [78], which could be related to the increase of arable lands in the catchment [75]. Current water quality monitoring programs are not able to identify the hydromorphological stressors existing in the area. A proper monitoring of these stressors should be designed, and adequate practices should be implemented to achieve a good ecological status.

## Supporting information

**S1 File.** Location of the sampling sites (Table A); **E**nvironmental variables studied at each site, and categories and codes used in the present study (Table B); Functional (biological and ecological) traits and categories studied according to Tachet et al. (2006) (Table C); Mean values (for four sampling dates) and ranges (i.e. minimum and maximum) of the physicochemical variables not included in Table 1 (Table D); Values of hydromorphological variables determined in the May sampling campaign and not included in Table 1 (components of IHF -River Habitat Index-, QBR -Riparian Forest Quality Index- and percentage of macrophytes cover) (Table E); Values of topographic, geologic and land cover variables not included in Table 1 (Table F); Density (ind/m$^2$) of macroinvertebrate taxa at each site (Table G); Relative abundance of each macroinvertebrate functional trait at study sites (Table H).
(DOCX)

## Author Contributions

**Conceptualization:** Rubén Ladrera, Narcís Prat, Miguel Cañedo-Argüelles.

**Data curation:** Rubén Ladrera.

**Formal analysis:** Rubén Ladrera, Oscar Belmar, Rafael Tomás, Miguel Cañedo-Argüelles.

**Funding acquisition:** Rubén Ladrera.

**Investigation:** Rubén Ladrera, Rafael Tomás.

**Methodology:** Rubén Ladrera, Oscar Belmar, Narcís Prat, Miguel Cañedo-Argüelles.

**Project administration:** Rubén Ladrera, Miguel Cañedo-Argüelles.

**Resources:** Rubén Ladrera, Oscar Belmar, Rafael Tomás.

**Software:** Rubén Ladrera.

**Supervision:** Rubén Ladrera, Miguel Cañedo-Argüelles.

**Validation:** Rubén Ladrera, Narcís Prat.

**Visualization:** Rubén Ladrera, Oscar Belmar, Miguel Cañedo-Argüelles.

**Writing – original draft:** Rubén Ladrera, Oscar Belmar, Miguel Cañedo-Argüelles.

**Writing – review & editing:** Rubén Ladrera, Oscar Belmar, Narcís Prat, Miguel Cañedo-Argüelles.

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
