## [Decision Letter · Decision Letter 0]

17 Jul 2019

PONE-D-19-15451

Agricultural impacts on streams near Nitrate Vulnerable Zones: a case study in the Ebro basin, northern Spain

PLOS ONE

Dear Dr. Cañedo-Argüelles Iglesias,

Thank you for submitting your manuscript to PLOS ONE. After careful consideration, we feel that it has merit but does not fully meet PLOS ONE’s publication criteria as it currently stands. Therefore, we invite you to submit a revised version of the manuscript that addresses the points raised during the review process.

Five reviewers have assessed your manuscript with detailed comments. Please carefully consider the concerns from Reviewer 1 about the statistical approach, the concerns about testing the initial hypothesis about the effect of nitrate (First comment Reviewer 3), and the concerns about the other putative stressors from Reviewer 3 (Comment 3).

We would appreciate receiving your revised manuscript by Aug 31 2019 11:59PM. To enhance the reproducibility of your results, we recommend that if applicable you deposit your laboratory protocols in protocols.io, where a protocol can be assigned its own identifier (DOI) such that it can be cited independently in the future. For instructions see: http://journals.plos.org/plosone/s/submission-guidelines#loc-laboratory-protocols

We look forward to receiving your revised manuscript.

Kind regards,

Clara Mendoza-Lera

Academic Editor

PLOS ONE

Journal Requirements:

1. In your Methods section, please provide additional location information of the study area, including geographic coordinates for the data set if available.

2. To comply with PLOS ONE submissions requirements for field studies, please provide the following information in the Methods section of the manuscript and in the “Ethics Statement” field of the submission form (via “Edit Submission”):

a) Provide the name of the authority who issued the permission for each location (for example, the authority responsible for a national park or other protected area of land or sea, the relevant regulatory body concerned with protection of wildlife, etc.). If the study was carried out on private land, please confirm that the owner of the land gave permission to conduct the study on this site.

b) For any locations/activities for which specific permission was not required, please

- i. state clearly that no specific permissions were required for these locations/activities, and provide details on why this is the case

- ii. confirm that the field studies did not involve endangered or protected species

c) For vertebrate studies only, please provide the following additional information:

- i. Full details of collection and sampling methods, including method of sacrifice if applicable

- ii. State whether the vertebrate work was approved by an Institutional Animal Care and Use Committee (IACUC) or equivalent animal ethics committee. If no approval was obtained, please explain why it was not required.

- iii. State clearly whether all sampling procedures and/or experimental manipulations were reviewed or specifically approved as part of obtaining the field permit.

For more information about PLOS ONE submissions requirements for field studies, please refer to http://journals.plos.org/plosone/s/submission-guidelines#loc-animal-research.

3. Thank you for including your funding statement; "This study was partially supported by the Institute of Studies of La Rioja"

Reviewers' comments:

Reviewer's Responses to Questions

**Comments to the Author**

1. Is the manuscript technically sound, and do the data support the conclusions?

Reviewer #1: Yes

Reviewer #2: Yes

Reviewer #3: Partly

Reviewer #4: Yes

Reviewer #5: Partly

2. Has the statistical analysis been performed appropriately and rigorously? 

Reviewer #1: No

Reviewer #2: Yes

Reviewer #3: Yes

Reviewer #4: No

Reviewer #5: Yes

3. Have the authors made all data underlying the findings in their manuscript fully available?

Reviewer #1: No

Reviewer #2: Yes

Reviewer #3: Yes

Reviewer #4: No

Reviewer #5: No

4. Is the manuscript presented in an intelligible fashion and written in standard English?

Reviewer #1: Yes

Reviewer #2: Yes

Reviewer #3: Yes

Reviewer #4: Yes

Reviewer #5: Yes

5. Review Comments to the Author

Reviewer #1: The manuscript “Agricultural impacts on streams near Nitrate Vulnerable Zones: a case study in the Ebro basin, northern Spain” analyses the taxonomic and trait composition of benthic invertebrate assemblages at 11 stream reaches in the Ebro river catchment. The studied stream reaches differ in pressure from agricultural activities in their catchments. The authors included in the analysis a long list of environmental variables, including water physical-chemical, hydromorphological, geological, topographic and land-cover descriptors. Results show that invertebrate assemblages differed between sites with low and high agricultural cover within their catchments, and that hydromorphological alterations have a prominent role in shaping biotic assemblages.

Overall, the manuscript is well structured, and figures, tables and supplementary materials are informative. I have some concerns regarding data analysis, in particular regarding the use of Mann-Whitney U test for the analysis of trait composition. Moreover, it is not written if the authors tested the explanatory variables for collinearity before running the DISTLM analysis, which is an important step for this kind of analyses. These and a few minor comments are detailed below.

The authors state that “All relevant data are within the manuscript and its Supporting Information files”, but I could not find the taxonomic and trait composition of benthic assemblages at the studied sites in the current submission.

Detailed comments:

Line 113: “we expected a shift…”: Specify whether the expected shift is along an environmental gradient or in time.

Line 124-127: How was the impact of urban/industrial/other anthropogenic pressures assessed? Did the assessment account for the area adjacent the stream or for the whole upstream catchment? What were the criteria for site selection? What was the size of the studied sites?

Line 144: “selective electrodes”: be more specific.

Lines 147-153: provide a brief description of the components of the QBR index and IHF index, and explain the scoring system (high values = high or low pressures?).

Lines 160 and 164: provide a brief description of the sampling and sub-sampling protocol.

Linea 163-164: What taxonomic level was used for Diptera, Gammaridae etc..?

Explain all acronyms in the text, e.g. IASPT and EPTCD at lines 173-174, DISTLM at line 187

Line 178: “microhabitat preferences”

Lines 186-196: Briefly explain what DISTLM analysis does. A large number of explanatory variables was included in the DISTLM. Collinearity among explanatory variables can affect the DISTLM analysis. Were environmental variables tested for collinearity?

Line 204: Although it is a non-parametric test, Mann-Whitney U test assumes that the distributions are similar. Figure 4 shows that in several cases the distributions of the two groups are not similar. I recommend choosing a more appropriate analysis. An alternative approach could be applying IndVal analysis to trait compositional data.

Line 236: Table 1 does not show among-date variability.

Line 266: It may be added here that the two groups separate sites with low and high agricultural land cover.

Lines 359-360: report your own results in this context.

Line 382: What does the expression “sediment increase” mean?

The grammar and language need a revision, here some examples:

Line 69: diverted instead of inversed?

Line 74: “river”

Line 180: delete ”agricultural”

Line 153: “the percentage”

Line 227, 334: “increased” implies a temporal dynamic, better “it was higher” or something like that.

Line 232: “Among the other”

Line 241-242: this sentence is not clear, I recommend rewording

Line 255: Change “determined” with “found”

Line 277: “referring to riparian vegetation”

Line 308: “shredders and scrapers feeding on”

Reviewer #2: This paper addresses the important issue of managing agricultural water pollution and how successful land management policies have been in improving the water environment. It is generally well written, lots of data are collected and these have been analysed thoroughly to provide some meaningful findings. I found very few problems with the paper and will just make a general comment about its scope which I feel should be addressed. I think the work done would benefit from being set in a more novel context. It is well known how agriculture impacts water quality and that nitrate pollution is driven by organic and inorganic fertiliser application as well as grazing livestock; this means that the first objective lacks novelty. The second objective is novel though and this could become the focus of the paper so that it becomes an assessment of the effectiveness of the Nitrates Directive and WFD in managing nitrate pollution, particularly in Mediterranean environments for which data are lacking. It also uses both physicochemical and macroinvertebrate data which many previous papers have not. Data are collected for many different variables and the lack of other activities, other than agriculture, in the study areas make this a useful assessment of the state of the environment in agricultural catchments. Some interesting data are presented which show high levels of pollution, particularly for the upstream areas of catchments. Indeed, it is concluded in lines 408-409 that NVZs and the WFD are not having an environmental impact, as has generally been found in the available literature, and this is a very important point. The authors need to state more clearly what is already known about the effectiveness of NVZs and include papers such as Howden et al. (2011), Kay et al. (2012), Weatherhead and Howden (2009), and Worrall et al. (2009).

Refs

Howden, N.J., Burt, T.P., Worrall, F. and Whelan, M.J., 2011. Monitoring fluvial water chemistry for trend detection: hydrological variability masks trends in datasets covering fewer than 12 years. Journal of Environmental Monitoring, 13(3), pp.514-521.

Kay, P., Grayson, R., Phillips, M., Stanley, K., Dodsworth, A., Hanson, A., Walker, A., Foulger, M., McDonnell, I. and Taylor, S., 2012. The effectiveness of agricultural stewardship for improving water quality at the catchment scale: experiences from an NVZ and ECSFDI watershed. Journal of Hydrology, 422, pp.10-16.

Weatherhead, E.K. and Howden, N.J.K., 2009. The relationship between land use and surface water resources in the UK. Land Use Policy, 26, pp.S243-S250.

Worrall, F., Spencer, E. and Burt, T.P., 2009. The effectiveness of nitrate vulnerable zones for limiting surface water nitrate concentrations. Journal of Hydrology, 370(1-4), pp.21-28.

Reviewer #3: This is a nice case-study highlighting that whilst nitrate is an important stressor of agricultural streams, other influence factors such as deposited fine sediment may be stronger contributors to changes in macroinvertebrate community structure and traits. The authors one-off sampled 11 streams in the upper Ebro basin for macroinvertebrates, and returned to the same sites four times over the span of a year to collect physicochemical data including water samples for the estimation of nitrate concentrations. The nitrate concentrations in a number of their sites are extremely high by global standards. They detected two site types by cluster analysis that reflected differences in the degree of agricultural impacts. Using constrained ordination, they associated certain environmental factors with differences in macroinvertebrate community structure that underpinned the two site types. The use of organismal traits highlighted that invertebrates in the degraded streams were potentially more tolerant of sedimentation (e.g., microhabitat, locomotion, food, feeding habitats).

The study is competently executed, the analyses appropriate, and the writing mostly sufficient. The study is hampered by the low site replication relative to the number of environmental variables measured. Some more mention should be made about underlying physical differences in the stream types (i.e., are they naturally different?). The Discussion could do with further work to tighten it up. Several questions remain unanswered that could be addressed by the authors. Below are my major scientific concerns, followed by more minor comments and suggested changes.

1. The conclusion that habitat degradation (e.g., sedimentation) is an important stress pathway from agricultural pressures rests largely on the results of the dbRDA and the trait analyses. However, I would like to see an explicit test of the key a priori hypothesis: namely, that nitrate is a major stress pathway in these streams (after all, nitrate is mentioned in the title). To do this explicitly, I would consider taking the constrained ordination further with variation partitioning to test if nitrate has a significant independent influence on macroinvertebrate community structure (see “varpart” in the “vegan” R package). Ideally, this analysis would also partition the other major sources of variation – which could include the other environmental factors (e.g., instream habitat) contributing to impairment, and spatial location (since degraded sites are generally closer together with some degree of non-independence). For an example of this approach I can recommend Burdon et al. 2013 (cited in the manuscript).

2. It would be good if the authors could highlight which of their environmental factors measured deposited sediment (i.e., yes, the IHF index tackles this but how?). At the moment, it is sort of inferred but not really explicit, which weakens the discussion.

3. There are other stressors and pressures that could contribute to changes that are not mentioned. Did the authors consider using the SPEAR index to assess potential pesticide impacts? Are there wastewater outfalls in the study basin that could contribute to the observed water quality issues? For example, site Z2 looks like it is downstream of a WWTP. For examples of studies assessing these dual pressures see:

Burdon, F. J., et al. (2019). "Agriculture versus wastewater pollution as drivers of macroinvertebrate community structure in streams." Science of The Total Environment 659: 1256-1265.

Burdon, F. J., et al. (2016). "Environmental context and magnitude of disturbance influence trait-mediated community responses to wastewater in streams." Ecology and Evolution 6(12): 3923–3939.

4. Pressure vs stressors: these terms are used interchangeably in the manuscript which I believe detracts from the message. If you refer to the Driver, Pressure, State, Impact, and Response (DPSIR) framework, “pressures” refer to broad categories contributing to impairment (i.e., land-use change, pollution, etc.). The multiple-stressor literature on agricultural systems then treats different pollutants as “stressors”, e.g., nutrients (nitrate), sediment etc. Thus, agriculture could be conceived as the pressure, with the key stressors as identified.

Minor comments:

1. Line 37: It would be good to hammer out a consistent approach to pressures and stressors (see above). Yes, there might be multiple “pressures” stemming from agricultural activities (e.g., water abstraction, pollution) – but here the focus is on agricultural stressors.

2. L39: pressures = stressors?

3. L49: the abbreviated i.e. should be followed by a comma. This needs to applied consistently throughout the MS.

4. L47-L51. As written this is confusing – we go from nitrate and different macroinvertebrate communities to then focusing on hydromorphological impacts. Perhaps the results from the dbRDA (and the analysis suggested at major pt.1 above) could be mentioned to help back this up?

5. L53. Consider replacing “consonance” with “concordance”.

6. L62. Now agriculture is a stressor. Consider making this the “pressure”, and everything else tested for the “stressors” (e.g., nutrients, etc.)

7. L63. Pesticides are agrochemicals. Consider using “fertilizers”

8. L65. Just write “pollutants” here.

9. L66-67. Reword. E.g. “…agriculture not only involves the heavy use of agrochemicals, but also requires large quantities of water for irrigation to maintain productivity.”

10. L67-68. “This demand for water has led to extensive modification of river systems in the past century, leading to greatly altered flow regimes.”

11. L70. Consider replacing “rectification” with “straightening”

12. L72. “There is an extensive published literature on the impacts of agricultural activities using macroinvertebrate assemblages as biological indicators, with some studies specifically addressing the impacts of nitrate concentrations.”

13. L74. “Nitrate impacts on river communite have been associated with other environmental factors, including increased autotrophic production that is exacerbated by losses in riparian vegetation due to reduced shading of the stream channel.”

14. L77. “…biomass greatly alters diurnal oxygen concentrations, with increased ecosystem respiration leading to impacts on pollution-sensitive taxa such as some EPT (REF). Excessive algal growth can also smother the riverbed, reducing substrate availability…”

15. L81. Delete “this”.

16. L85. Replace “They” with “Sedimentation also alters…

17. L86-87. This pt. about functional traits feels shoehorned in. Perhaps first make the point about functional changes in response to sedimentation with the specific responses (L87-M89). Then more broadly describe the use of traits to discriminate agricultural stressors in impacted streams for the last sentence in the paragraph.

18. L90. “directives (i.e., a legal act requiring member states to achieve a certain result)”

19. L91. “revert” = “reverse”

20. L96. “recover” = “restore”

21. L100. “…impacts at the basin…”

22. L104. “pointed” = “singled”. “pressure” = “stressor”

23. L107. This is correct use of pressures. Are there wastewater treatment plants in the catchment?

24. L107. “…because it is subjected to considerable human pressures, including agriculture.”

25. L108. “pressure” = “stressor”

26. L113-114. Order – “taxonomic composition and functional structure”

27. L115. “pollution-sensitive”

28. L116. Better to say “leaf litter” for “vegetal detritus”

29. L131. “conserved” = “preserved”

30. L132. “scarce” = “limited”

31. L136. “(i.e., February, May, August, and November)” – note use of Oxford comma after August.

32. L138. “exclusively” = “only”

33. L141. “studied” = “recorded”?

34. L144. Change “selective electrodes” to “handheld probes” – the Make and Model, etc. should be listed here

35. L160-161. Write a brief description of the sampling method for the non-Spanish speaking readers.

36. L164. Check spelling of “Hydrachnidia”

37. L173-174. Need to explain what EPT, IASPT, and EPTCD acronyms stand for. Are there references for these indices?

38. L182-184. Doesn´t lines 181-182 describe the fuzzy coding approach?

39. L188. “genera abundances”

40. L191. “physicochemical”

41. L215. “August”

42. L216. Table 1. The sites could be ordered by their groups (i.e., 1 or 2). Are those nitrate concentrations for real!? Wow. Did the authors measure salinity too – the conductivities are extremely high too?

43. L226. “conductivity dropped below 500…”

44. L229-230. Put site abbreviation in parentheses (Enc)

45. L230-231. Is salinity an issue here?

46. L240. I would highlight what the IHF index measures i.e., “Instream habitat, as measured by the IHF index, differed…”

47. L247. Likewise, remind the readers what the QBR is about.

48. L262. I word rephrase this sub-heading “Macroinvertebrate assemblages and agricultural stressors”

49. L272. Use “Instream habitat (IHF index; 43.9%)”. Need to explain what this is showing (i.e., less sediment, more heterogeneous substrate, etc.)

50. L274. Component of the “riparian habitat index (QBR) referring….”

51. L276. “…to the explained model variance.”

52. L278. Is salinity as issue if chloride is prominent?

53. L289. These are Families – can the authors name some common EPT species or genera? E.g., in Table 2. Rithrogena is mentioned.

54. L319. This is a very weak way to start the Discussion. Better to say “Our results indicate that stressors associated with agricultural activities have measurable influences on macroinvertebrate assemblages in the study region.”

55. L321. This result is not confounded by the presence of wastewater?

56. L325. Writing: “Concentrations of nitrate significantly….”

57. L328. Hyphenate “oxygen-carrying”

58. L336-337. I know this aspect is mentioned later, but it might good to integrate the trait findings the measured stressors (e.g., relate the coverage of macrophytes to the traits response, and then how benthic habitat changed driving microhabitat preferences etc.). I.e. make some broad statements about the results and the key hypothesis: i.e., nitrate is a major stressor of concern, before discussing the specific results.

59. L356-359. This sentence needs work

60. L401. There are many studies that have successfully used traits to describe agricultural impacts. Consider citing a few more of these, and rewording “In agreement with other studies (REFS).”

61. L405-407. This is an important finding, and I think you really draw it out with a variation partitioning analysis (i.e., explicitly testing the independent contribution of nitrate relative to the other influence factors).

Reviewer #4: This manuscript aims at identifying the main pressures on stream macroinvertebrates in relation to intensive agriculture around the so-called Nitrate Valuable Zones. The authors collected a rather impressive amount of “background” data (i.e. environmental data) as well as taxonomic and functional composition of the macroinvertebrate community. Although I find the research questions that this manuscript aims at highlighting, I believe it needs some profound revision including new statistical analyses. Anyhow, this study shows that channel simplification, riparian forest degradation and sediment inputs were the main factors affecting invertebrate assemblages pointing towards the need for implementing management practices not exclusively focused on nitrate reduction.

I have some comments that I believe could help the authors to improve the manuscript.

Main points:

1. The introduction reads quite well: the objectives and the hypotheses that are going to be tested are clearly stated. However, several topics of prime importance – in my opinion – were eluded such as the complementarity between taxonomic and functional information, the importance of traits, the reasons why taxonomic composition is so used, etc. (see Truchy et al, Adv. Ecol Res 53, 55-96 for an overview and trait classification). Furthermore, I felt that the importance of the research question addressed here was lifted too late in the introduction (in the second to last paragraph) and that the transitions between the different paragraphs/ideas were abrupt.

2. The method section needs clearer justification for such sampling design (i.e. several sampling sites per stream, meaning the downstream sites are potentially capturing the effects already captured by the most upstream sites…). Also I have some concerns regarding the trait classification used by the authors. Although the authors seem to comply with the classification developed by Usseglio-Polatera et al., 2000 (Freshwater Biol. 43, 175-205) that separates traits into biological vs. ecological traits, some of the traits used in this study are designated as biological traits but fall in reality in the ecological trait category (e.g. trophic status). I would rather recommend the authors to use another approach and consider response vs. effect traits as I think the hypotheses and therefore the conclusion of the study could be broaden to ecosystem functioning. In this respect, using traits belonging to the category of effect traits would definitely strengthen the message of the study. And least but not last, I think some complementary analyses are needed (maybe those were already conducted but then they should have been in the supplementary information). I am not very familiar with the statistical approach used by the authors (i.e. DISTLM) and therefore some justification regarding the choice of such statistics rather than the more “standard” set of analysis would have been appreciated (see my specific comments). I would also have liked to see 1) a PCA on the environmental variables rather than this very long and wordy description of the environmental characteristics of the sites in the result section; 2) a multivariate analysis on the species traits to observe the results in a multidimentional space and see which environmental variables trigger which trait (e.g. PCA, see Frainer et al. 2018, JAE 55: 377-385 for examples of analyses).

3. For the result section, the first section could be easily cleared up by removing all the unnecessary acronyms and site names and focus only on the environmental variables that the authors do discuss later. Also, a multivariate analysis on those environmental data would actually be an efficient way to summarize the information. Also, there is still some confusion around the DISTLM analysis as the authors seem to present a dbRDA in Figure 3. Finally, I would also suggest the authors to run a multivariate analysis on the trait data as already mentioned.

4. The discussion is overall well written and broaden quite nicely on recommendations for managers. But it would gain in strength if the authors started by making a “summary” paragraph where they would highlight their main results and whether or not their initial hypotheses were supported. By using response vs. effect traits, the authors could also extrapolate their results to stream ecosystem functioning, something that they just touch upon. This would definitively add some interest to the discussion.

Specific comments:

L39-40: the main pressures on what?

L60-71: too many “the most”. This paragraph is very long and could easily get shortened for a more efficient introduction.

L156: Layers of what?

L177: First time you mention biological traits: What are those? What kind of information can they give you? It would be good to already introduce the concept of traits in the introduction.

L178-184: No justification at all regarding the inclusion of these specific traits. Why did you choose those? How do you expect them to be affected by nitrate pollution? A reference to Usseglio-Polatera et al.’s work would be appreciated here.

L187: What does DISTLM stand for? And what does it do? I guess the standard procedure would have been an NMDS followed by a PERMANOVA potentially complemented with a BEST procedure.

L191: Should be physico-chemical

L205: Differences in what?

L215: Should be “in August”

Table 1: It would be good to specify again the meaning of the acronyms as the table and its legend should be able to stand alone.

L264-266: What are the main differences between those 2 groups? Agriculture vs. forest? Or something else?

L275-279: If not significant, why do you include those variables? It seems to be a little bit odd as you conducted the DISTLM for selecting the most important environmental variables driving differences in macroinvertebrate assemblages and yet you include variables that do not participate significantly in explaining differences…

Figure 3: Seems to be a dbRDA. dbRDA is not mentioned at all in the methods section and the caption of the figure refers to DISTLM analysis. So please, be more precise in the description of the methods/figures.

L289-295: I would suggest the authors to shift the order of this paragraph with the one L270-279 as it would be more natural saying that 2 groups were identified and what they consist in.

L307: The trait categories differ in terms of what? Means? Variance? Both?

Figure 4: It is very hard to distinguish the white lines unfortunately. I would also recommend the authors to draw asterisks when the differences between the means are significant.

L323: Do the authors really mean resilient strategies? Or resistance strategies? See Bogan et al, 2015, Freshwater Biol. 60, 2547-2558 for definitions.

L336-337: This sentence reads a little bit weird. It would be good if the authors could revise it.

Supplementary:

Table S4: Please expand the legend. What are those values? Means? What are the acronyms for?

Reviewer #5: The study and analyses were conducted well and contribute to regional and general knowledge of agricultural influences to stream benthic macroinvertebrates (especially as indicated by their traits). My only major concern is that the conclusions and recommendations as written are beyond the scope of the study (see General Comment 1).

I recommend minor revisions be made to the manuscript.

*** General Comments ***

1) The conclusions and recommendations are far outside the scope of, and thus unsupported by, the evidence presented here. Please constrain the conclusions and recommendations to those justifiable directly from the evidence. Your data and analyses do not address the efficacy of policies or practices (e.g., “dredging and channelization restrictions”) in achieving a reduction in the deleterious factors you have identified; it is possible there is are better strategies to achieve the desired beneficial outcome. Your data can only support the conclusion that less habitat alteration may improve ecological status.

2) How was macroinvertebrate sampling effort (to include enumeration effort) standardized across samples? Were adjustments made during statistical analysis where appropriate to account for any influence of variable sample effort? The macroinvertebrate enumeration protocol described in the manuscript appears to allow for a variable level of effort in enumeration (I trust that sample collection effort per Spanish protocol is standardized among sites; if not, that is of concern as well and should be addressed). As written, the manuscript relates an unclear and potentially inconsistent sub-sampling approach that could yield samples with vastly different number of individuals (level of effort) per sample. As written, the text suggests that some samples may have been sub-sampled, some may not have been, and the criteria are not clearly presented that were used for deciding when to subsample and when to stop counting after 300 individuals. As greater effort would naturally yield greater taxa richness (with implications for statistical analysis and interpretation), it is important to ensure that any observed increase in richness is not an artifact of the sampling procedure.

3) Results section could be improved to better convey synthesized findings, as opposed to summary of measurements (which are in the Tables). By that I mean, tell the reader what you found; what should the reader take away after reading Results? To me, this section would be more compelling if it focused on conveying the findings that certain abiotic factors differed between biologically-different groups 1 and 2.

Additionally, the section could benefit from a bit more context; frame values relative to what is expected in streams of the region with minimal impact so the reader understands whether those values are of concern.

As well, or at the least, I suggest rearranging all presentation of data in tables, and possibly in text, to be grouped by the site groupings that were developed through cluster analysis (as opposed to the alphabetical arrangement at present). As those groupings are used for Fig 3 and Fig 4, seeing the abiotic data in tables grouped accordingly should facilitate easier contemplation of associations between abiotic and biotic data. For example, it is notable that, with a few exceptions, sites in Group 1 have lower specific conductance than sites in Group 2. Such an adjustment could also work well to more clearly convey differences between groups as presented in Results L211-261 and make the entire section more compelling. Given that you have established that biological differences exist between groups (Fig 2, Fig 3), presenting the abiotic information accordingly is more ecologically meaningful than site abbreviations that mean nothing to the unacquainted reader.

4) Regarding averaging physicochemical parameters (L191-193, L225, and Tables 1 and S3), I generally agree that an annual mean derived from quarterly sampling can give a better long-term or overall representation of conditions to which organisms are exposed as compared to just one season of data. However, such averaging can mask seasonally-variable parameters that may have different effects at different seasons.

Summer, for example, is a particularly stressful time of year as it combines heat stress (summer maximum temperature) with low dissolved oxygen saturation potential (resulting from said heat). Similarly, nutrient levels are potentially of greater influence during the growing season (or leaf-off season) when sunlight and/or temperature are at levels sufficient to induce rapid excessive algal growth. Further, presenting the standard error of the mean in Table 1 is of marginal utility in describing values that vary widely across seasons (as noted with nitrate on L236-237). I’d prefer to see mean and range, the latter of which is helpful for understanding if extreme values (e.g., low summer oxygen, high summer temperature) may be influencing biota.

Another option to consider is to present these values in figure format, with 2 or 3 Y-axes (or multiple panels), and with 4 seasonal points along the X-axis. For one, it would allow graphical representation of thresholds and whether they were exceeded (with a multipanel presentation), which is somewhat more interesting than the absolute concentrations in the context of this manuscript. Secondly, visual separation of sites by Group would be evident where it exists, which could help with regard to the concern noted in General Comment 3. Presentation of seasonal values would also allow for contemplation of the timing of chemical levels relative to taxa life stages; something potentially of interest as there appears to be some differences among groups with respect to reproductive cycles (Fig 4). The numerical table may still reside in Supporting Information.

*** Specific Comments ***

L76-77: “High periphyton biomass reduces diurnal oxygen concentrations” is confusing. Diurnal means “during the day”, at which time periphyton is photosynthesizing and producing oxygen. In contrast, at night (nocturnal) respiration by excessive periphyton could cause substantial decrease in oxygen. A quick look at the citation (#22) does not clarify to what those authors were attributing the low diurnal oxygen. Moreover, the statement in the citation restricts the comment about low oxygen to substrate interstices, yet the language of the present manuscript is not specific and could reasonbly be interpreted to mean water-column oxygen concentration is reduced. It’s possible that the Harding et al. paper was attempting to state that diel (i.e., during a 24-h period) oxygen was depressed owing to excessive nocturnal respiration. Regardless, a revision for clarity and accuracy (e.g., interstices vs. water-column) would help here, as it is important to establish clearly the mechanisms by which algal growth influences macroinvertebrates since that is one of several important factors as noted on L81-82.

L111-117: This is close, but not quite a true statement of hypotheses; they are only predictions. Hypotheses should suggest mechanisms by which the expected effects are thought to occur. The authors cover that ground adequately in the prior paragraphs discussing mechanisms by which each of the factors on L111-113 (nitrate, sediment, habitat) are expected to influence macroinvertebrates. Revising the phrasing of L111-113 to include such mechanistic statements would present a true hypothesis, with related predicitons following on L113-117.

L147-154 (and similarly L120-127): A brief description of sample reach length and habitat sampling methods (e.g., transects?) should be provided.

L151-153: Terms used to refer to habitat features in the text differ from those used in Table S1 (e.g., rapid vs. riffle). Terminology should be consistent throughout the manuscript.

L160-161: Please provide a brief summary of general sampling methods (e.g., quantitative vs. semi-quantitative, sample device, approx. substrate area sampled, substrate type(s) sampled, etc.). This clarification will also help address the concerns over standardized sampling effort that I raised above.

L164-165: “At least 300 individuals per sample…” Was adjustment made for level of effort (e.g., greater taxa richness accumulation with greater number of individuals)? Or was it accounted for within the calculations of the various indices used or in some other way (e.g., non-subsampled sample data rarified to a number of individuals equal to the lowest abundance among samples)? Please clarify.

L167-176: Many abbreviations are used in this section without prior definition. Please define.

L216-223 (Table 1): Please choose a different convention from red text to indicate notable values, as that may be illegible to color-blind readers and when reproduced in black & white. Italics could work.

L216-223 (Table 1): For the habitat indices (IHF and QBR), please add perspective as to what are generally good vs. poor values (or comparable to reference vs. increasingly different from reference) here in the table and ideally also in the Methods section (L147-154; authors describe the indices but don’t provide orientation to the range of values or what they mean).

L231-232: Some perspective would be helpful here. It is stated that sulfate and calcium were “exceptionally high” at Encemero. For the unacquainted reader, it would be good to state what an expected levels of these parameters are for reference streams of the region. Or at least contrast with and note values in the least impacted streams in this study.

L242-245: The sentence is awkward and unclear. The numbers in parentheses are unnecessary and confusing (I initially thought they were the low values alluded to at the beginning of the sentence). Please revise for clarity. Further, this is a prime location to highlight Group-wise differences.

L270 & Fig 3 caption: Clarify that DISTLM analysis is explaining variation in macroinvertebrate assemblage composition among sites.

L275-276: Please indicate Type I error (alpha) levels for significance determination.

L319-321: Language here is too conclusory regarding causation for an observational study. Please restate to reflect the correlative nature of the study and analyses.

L325: Causation; see above

L326-327: Causation; see above

L334-336: Causation; see above

L358: Please clarify what is meant by “elements heterogeneity”.

L362: Causation; see above

L401: Causation; see above

L405: On what basis has sedimentation been parsed from other factors in this study as a main driver of degradation? The largest proportion of variation was noted as the aggregate IHF-total, and no sediment-specific parameters are included in the list of other significant explanatory factors (L270-279). Presumably sediment components of the IHF were included in DISTLM just as was done with QBR-natur, yet those IHF components were not prominent factors differentiating groups. I don’t think the data and analyses here support such a narrow conclusion as written.

6. PLOS authors have the option to publish the peer review history of their article (what does this mean?). If published, this will include your full peer review and any attached files.

Reviewer #1: No

Reviewer #2: Yes: Paul Kay

Reviewer #3: No

Reviewer #4: No

Reviewer #5: No

---

## [Author Response · Author response to Decision Letter 0]

23 Sep 2019

Dear Editor, 

First of all, we would like to thank you and the reviewers for the valuable contributions to the manuscript, which have considerably improved it. We have included new tables in the Supporting Information and modified tables and figures in the manuscript according to the reviewer´s comments. We have also modified the structure of the different sections (i.e. rearranged them) and the content of the manuscript (e.g. by providing additional information to clarify some issues). 

Below we provide detailed answers to every major comment and to minor ones when necessary (some, such as minor text modifications, have been incorporated without further discussion).

REVIEWER 1

MAJOR COMMENTS

1. Reviewer comment: Regarding the use of Mann-Whitney U test for the analysis of trait composition. Although it is a non-parametric test, Mann-Whitney U test assumes that the distributions are similar. Figure 4 shows that in several cases the distributions of the two groups are not similar. I recommend choosing a more appropriate analysis. An alternative approach could be applying IndVal analysis to trait compositional data.

Response: Mann-Whitney U test assumes that, under the null hypothesis H0, the distributions of both groups are equal, and under the alternative hypothesis H1 the distributions are not equal. Therefore, we consider that this test is appropriate to determine functional differences between groups. Nevertheless, following the reviewer’s suggestion, we have performed an IndVal analysis. The results are shown below: 

Trait category group indval pvalue Trait category group indval pvalue

Fedding habits: deposit_feeder 2 0,659 0,002 Microhabitat: bould_cobbl_peb 1 0,569 0,029

Fedding habits: filter_feeder 2 0,620 0,028 Microhabitat: gravels 1 0,542 0,014

Fedding habits: parasite 2 0,673 0,002 Microhabitat: macrophytes 2 0,588 0,018

Fedding habits: scraper 1 0,560 0,012 Microhabitat: mud 2 0,622 0,004

Fedding habits: shredder 1 0,624 0,008 Reprod. Cycles: 1 1 0,589 0,004

Food: detritus_less1 2 0,564 0,007 Reprod. Cycles: less_1 1 0,674 0,006

Food: fine_sediment 2 0,627 0,008 Reprod. Cycles: more_1 2 0,642 0,003

Food: macrophytes 2 0,544 0,006 Respiration: gill 1 0,570 0,009

Food: microphytes 1 0,541 0,005 Respiration: plastron 1 0,775 0,002

Locomotion: burrower 2 0,578 0,008 Respiration: tegument 2 0,560 0,037

Locomotion: crawler 1 0,553 0,003 Trophic status: eutrophic 2 0,635 0,003

Locomotion: flier 1 0,688 0,011 Trophic status: mesotrophic 2 0,526 0,009

Locomotion: swimmer 2 0,550 0,039 Trophic status: oligotrophic 1 0,602 0,003

A total of 26 functional traits were significantly associated to one of the two groups, 25 of which were also significantly different between group 1 and 2 according to the Mann-Whitney U test. Tegument respiration was the only trait selected by the IndVal analysis that did not differ significantly between groups according to the Mann-Whitney U test.

Since the results from the two analyses are very similar, we have decided to keep the Mann-Whitney U test in the manuscript. With only two groups, we were not interested in assigning an indicator value to each trait characteristic. Besides, as the groups were defined using macroinvertebrate composition, we consider that applying the Indval again on trait characteristics (as we have done with composition) could be redundant. Moreover, we think that the “beanplots” are very illustrative of the differences in trait characteristics (supported by the Mann-Whitney tests).

2. Reviewer comment: It is not written if the authors tested the explanatory variables for collinearity before running the DISTLM analysis, which is an important step for this kind of analyses. 

Response: We have explained the DISTLM method in greater detail in the new version of the manuscript, including aspects related to the collinearity of variables. Variables highly correlated with others (i.e. Spearman correlation coefficient higher than 0.9) had been previously removed from the analysis.

3. Reviewer comment: The authors state that “All relevant data are within the manuscript and its Supporting Information files”, but I could not find the taxonomic and trait composition of benthic assemblages at the studied sites in the current submission.

Response: The taxonomic and trait composition of the macroinvertebrate assemblages at the studied sites have been included in the Supporting Information.

MINOR COMMENTS

1. Reviewer comment: How was the impact of urban/industrial/other anthropogenic pressures assessed? Did the assessment account for the area adjacent the stream or for the whole upstream catchment? What were the criteria for site selection? What was the size of the studied sites?

Response: As we have detailed in the revised manuscript, to reduce natural differences among the studied sites (100 m length), all of them belong to first-order streams and to the same stream typology ("Calcareous Mediterranean mountain rivers") according to the classification developed in Spain by the Ministry for the Environment to fulfil the WFD (ORDER ARM/2656/2008). The study sites were not affected by urban or industrial discharges. There are no large villages (i.e. more than 500 inhabitants) in the area and almost no industrial activity (agriculture is the main activity). The study sites were selected to fulfil the above-mentioned criteria after a visual inspection of the region.

REVIEWER 2

MAJOR COMMENTS

1. Reviewer comment: Some interesting data are presented which show high levels of pollution, particularly for the upstream areas of catchments. Indeed, it is concluded in lines 408-409 that NVZs and the WFD are not having an environmental impact, as has generally been found in the available literature, and this is a very important point. The authors need to state more clearly what is already known about the effectiveness of NVZs and include papers such as Howden et al. (2011), Kay et al. (2012), Weatherhead and Howden (2009), and Worrall et al. (2009).

Response: We thank the reviewer for the recommended literature. We have revised it and included it in the “conclusion and recommendation” section to provide more information on the Nitrate Directive and the NVZs effectiveness, and to expand on the management implications of our study.

REVIEWER 3

MAJOR COMMENTS

1. Reviewer comment: The conclusion that habitat degradation (e.g., sedimentation) is an important stress pathway from agricultural pressures rests largely on the results of the dbRDA and the trait analyses. However, I would like to see an explicit test of the key a priori hypothesis: namely, that nitrate is a major stress pathway in these streams (after all, nitrate is mentioned in the title). To do this explicitly, I would consider taking the constrained ordination further with variation partitioning to test if nitrate has a significant independent influence on macroinvertebrate community structure (see “varpart” in the “vegan” R package). 

Response: We have revised the text to better explain the use of DISTLM to determine the importance of the different environmental variables in the variation of the macroinvertebrate assemblages among sites. “Varpart” (vegan) partitions variance using a Redundancy Analysis, whereas DISTLM does it using a Distance-based Redundancy Analysis. The DISTLM analysis allows analysing and modelling the relationship between a biological data cloud (i.e. a composition resemblance matrix) and one or more predictor variables. In our case, the resemblance matrix described the dissimilarities among the macroinvertebrate assemblages of the studied sites based on genus abundance data, and we used the environmental variables (including nitrate) as predictor variables. The DISTLM provided quantitative measures of the variation explained by each predictor, allowing the partitioning of variance according to a regression model.

As it is described in the new version of the manuscript, according to DISTLM analysis, nitrate significantly explained a 9.3% of the variation in the macroinvertebrate assemblages between sites. However, the variation in aquatic macroinvertebrate assemblages among sites was best explained by the variables associated with hydromorphological alterations of the stream habitat and topography. Specifically, the combination of the IHF index, the QBR components that refer to riparian vegetation cover and river channel alteration, and the ratio of valley width to channel width (VWI) explained a 67% of the variation in the macroinvertebrate assemblage among sites.

2. Reviewer comment: It would be good if the authors could highlight which of their environmental factors measured deposited sediment (i.e., yes, the IHF index tackles this but how?). At the moment, it is sort of inferred but not really explicit, which weakens the discussion.

Response: Deposited sediment is directly related to the component 1 of the IHF index. As we have detailed in the new version of the manuscript and in the Supporting Information, this component refers to hard substrate embeddedness into the fine sediment. Table S4 shows that every site in group 2 has a score of 0 in this component, which means that more than 60% of the hard substrate was embedded in fine sediment, whereas every site of group 1 showed a higher score (i.e. lower embeddedness).

3. Reviewer comment: There are other stressors and pressures that could contribute to changes that are not mentioned. Did the authors consider using the SPEAR index to assess potential pesticide impacts? Are there wastewater outfalls in the study basin that could contribute to the observed water quality issues? For example, site Z2 looks like it is downstream of a WWTP. 

Response: Regarding other stressors (e.g. pesticides) and indices (e.g. SPEAR) not included in the study, we would like to highlight that the Ebro Hydrographic Confederation (the public organism responsible of water quality monitoring programmes in the rivers of the area; http://www.chebro.es/) routinely analyses pesticides (together with nutrients and other chemical compounds) in the study region. Among them, nitrate has been historically pointed out as the main pollutant and stressor of these water bodies (public data on http://www.chebro.es/). Thus, although we recognise that it could be interesting to include pesticides in our study, we decided to focus on nitrate. Along with nutrients, we decided to assess hydromorphological variables because they have been pointed out as key factors shaping aquatic invertebrate communities in agricultural streams by previous papers and they have received little attention from the monitoring programmes. 

Regarding wastewater treatment plants, we selected sites with no urban or industrial sewage discharges in their adjacent area. It should also be noted that there are no large villages (i.e. more than 500 inhabitants) in the area and almost no industrial activity (agriculture is the main activity). Z2 (belonging to Zamaca stream) is the closest site to a wastewater treatment plant, and it is located more than 2 km upstream of the sample site. Moreover, the wastewater plant is dimensioned to 3500 inhabitants (https://www.larioja.org/larioja-client/cm/consorcio-aguas/images?idMmedia=892956), and it collects discharges from 3 villages (Zarratón, Ollauri and Gimileo) that have 660 inhabitants in total (data from Spanish National Statistics Institute; INE). Thus, it guarantees an important reduction of pollutants before the effluent reaches Zamaca stream. For example, in 2017 nitrate concentration in the effluent of this wastewater plant was always lower than 18 ppm (https://www.larioja.org/consorcio-aguas/es/depuracion/instalaciones/depuradoras-servicio/estaciones-depuradoras-aguas-residuales), whereas the average nitrate concentration at Z2 was 84 ppm. Thus, the influence of the wastewater treatment plant seems to be very limited. 

4. Reviewer comment: Pressure vs stressors: these terms are used interchangeably in the manuscript which I believe detracts from the message. If you refer to the Driver, Pressure, State, Impact, and Response (DPSIR) framework, “pressures” refer to broad categories contributing to impairment (i.e., land-use change, pollution, etc.). The multiple-stressor literature on agricultural systems then treats different pollutants as “stressors”, e.g., nutrients (nitrate), sediment etc. Thus, agriculture could be conceived as the pressure, with the key stressors as identified.

Response: The terms pressures and stressors have been corrected following the reviewer´s suggestions.

MINOR COMMENTS

1. Reviewer comment: Table 1. The sites could be ordered by their groups (i.e., 1 or 2). Are those nitrate concentrations for real!? Wow. Did the authors measure salinity too – the conductivities are extremely high too?

Response: Following the reviewer’s suggestion, the sites have been ordered by group in the tables of the manuscript and in the Supporting Information. Relative to high conductivity values, as we have added to the manuscript, the high conductivity and concentrations of different ions at Enc and R3 sites are associated with natural sodium sulfate tertiary deposits and secondary minerals, mainly gypsum and calcite (Menduiña et al. 1984)

Menduiña J, Ordóñez S, del Cura MG. Geología del yacimiento de glauberita de Cerezo de Rio Tirón (Burgos). Bol Geol Min. 1984;95(1):35–51.

REVIEWER 4

MAJOR COMMENTS

1. Reviewer comment: The introduction reads quite well: the objectives and the hypotheses that are going to be tested are clearly stated. However, several topics of prime importance – in my opinion – were eluded such as the complementarity between taxonomic and functional information, the importance of traits, the reasons why taxonomic composition is so used, etc. (see Truchy et al, Adv. Ecol Res 53, 55-96 for an overview and trait classification). Furthermore, I felt that the importance of the research question addressed here was lifted too late in the introduction (in the second to last paragraph) and that the transitions between the different paragraphs/ideas were abrupt. 

L60-71: too many “the most”. This paragraph is very long and could easily get shortened for a more efficient introduction.

Response: We thank the reviewer for the valuable suggestion and reference. We have rewritten the introduction, reduced the first paragraph and explained the concept of biological traits and its use in biomonitoring.

2. Reviewer comment: The method section needs clearer justification for such sampling design (i.e. several sampling sites per stream, meaning the downstream sites are potentially capturing the effects already captured by the most upstream sites…). Also I have some concerns regarding the trait classification used by the authors. Although the authors seem to comply with the classification developed by Usseglio-Polatera et al., 2000 (Freshwater Biol. 43, 175-205) that separates traits into biological vs. ecological traits, some of the traits used in this study are designated as biological traits but fall in reality in the ecological trait category (e.g. trophic status). I would rather recommend the authors to use another approach and consider response vs. effect traits as I think the hypotheses and therefore the conclusion of the study could be broaden to ecosystem functioning. In this respect, using traits belonging to the category of effect traits would definitely strengthen the message of the study. And least but not last, I think some complementary analyses are needed (maybe those were already conducted but then they should have been in the supplementary information). I am not very familiar with the statistical approach used by the authors (i.e. DISTLM) and therefore some justification regarding the choice of such statistics rather than the more “standard” set of analysis would have been appreciated (see my specific comments). I would also have liked to see 1) a PCA on the environmental variables rather than this very long and wordy description of the environmental characteristics of the sites in the result section; 2) a multivariate analysis on the species traits to observe the results in a multidimentional space and see which environmental variables trigger which trait (e.g. PCA, see Frainer et al. 2018, JAE 55: 377-385 for examples of analyses). What does DISTLM stand for? And what does it do? I guess the standard procedure would have been an NMDS followed by a PERMANOVA potentially complemented with a BEST procedure.

Response: The sampling design has been explained and justified in the methods section of the manuscript. The 6 streams selected in the present study are the only first-order streams around these NVZs, where nitrate levels have previously exceeded, according to monitoring programs carried out by the Ebro Hydrographic Confederation (CHE; http://www.chebro.es/), the highest value established by WFD to consider that a river is in good ecological status. However, some of these streams have been studied just sporadically and all of them just at one site. Since the intensity of agricultural stressors may vary at reach scale, we included additional sampling sites.

We have now divided the functional traits into biological and ecological ones in accordance to Usseglio-Polatera et al., 2000, explaining the meaning of each one in the methodology and modifying Figure 4. We believe that this approach is suitable for our study, but we have also made a brief reference to the concept of effect traits both in methodology and in discussion. Finally, we have justified our selection of traits.

Usseglio-Polatera P, Bournaud M, Richoux P, Tachet H. Biological and ecological traits of benthic freshwater macroinvertebrates: Relationships and definition of groups with similar traits. Freshw Biol. 2000;43(2):175–205.

According to the reviewer’s comment, DISTLM analysis routine and its dbRDA associated plot have been better explained in the new version of the manuscript. We believe that no further analyses are required to meet our objectives, since the current analyses already quantify the response of the macroinvertebrate communities to shifts in the different predictor variables and identify the main traits that are affected by agricultural impacts. According to the manual of the PERMANOVA+ software, DISTLM is a more precise analogue of multiple linear regression than BEST, as it uses the resemblance matrix as an active sheet.

Frainier et al. (2018) used a PCA to summarise the variables in components that were then used in further analyses. This is an appropriate procedure to summarize information when the use of too many variables can difficult the observation of patterns. However, the observed relationships between two specific variables depend largely on the set of variables (adding or deleting variables may alter the apparent relationships among them) and the number of dimensions considered to represent the results. As we were not interested in summarising variables into components, we used the DISTLM to observe relationships between biological and environmental variables (without the influence of the set of variables or number of dimensions considered). The differences in functional characteristics are determined by the differences in composition between the two groups of sites defined, which are shaped by the different environmental conditions. These functional differences are illustrated using beanplots (statistically tested using Mann-Whitney).

3. Reviewer comment: For the result section, the first section could be easily cleared up by removing all the unnecessary acronyms and site names and focus only on the environmental variables that the authors do discuss later. Also, a multivariate analysis on those environmental data would actually be an efficient way to summarize the information. Also, there is still some confusion around the DISTLM analysis as the authors seem to present a dbRDA in Figure 3. Finally, I would also suggest the authors to run a multivariate analysis on the trait data as already mentioned.

Response: According to the reviewer’s comments, we have synthesized the environmental variables, focusing on those that are later discussed. Also, we have grouped the sites in the tables according to the cluster analysis to make the results easier to follow. We consider necessary to include these tables, since they show the physicochemical characterisation of the sites and their ecological status according to the WFD, which was one of the objectives of the study. Regarding the PCA analysis, please see our reply above.

4. Reviewer comment: The discussion is overall well written and broaden quite nicely on recommendations for managers. But it would gain in strength if the authors started by making a “summary” paragraph where they would highlight their main results and whether or not their initial hypotheses were supported. By using response vs. effect traits, the authors could also extrapolate their results to stream ecosystem functioning, something that they just touch upon. This would definitively add some interest to the discussion.

Response: According to the reviewer’s comments, we have added a “summary” paragraph to the discussion section where the main results and their relationship with our initial hypotheses are summarised. We have also made a brief reference to the concept of effect traits both in methodology and in discussion

MINOR COMMENTS

1. Reviewer comment: Figure 4, it is very hard to distinguish the white lines unfortunately. I would also recommend the authors to draw asterisks when the differences between the means are significant.

Response: Figure 4 has been modified according to the trait classification -biological or ecological- and it has been redrawn to make lines more visible. We did not draw asterisks as the figure only includes trait categories that were significantly different between groups (this is explained in the figure caption).

REVIEWER 5

MAJOR COMMENTS

1. Reviewer comment: The conclusions and recommendations are far outside the scope of, and thus unsupported by, the evidence presented here. Please constrain the conclusions and recommendations to those justifiable directly from the evidence. Your data and analyses do not address the efficacy of policies or practices (e.g., “dredging and channelization restrictions”) in achieving a reduction in the deleterious factors you have identified; it is possible there is are better strategies to achieve the desired beneficial outcome. Your data can only support the conclusion that less habitat alteration may improve ecological status.

We have reworded the conclusions and recommendations to avoid overstatements and stick to the evidences supported by our results. Those addressing the efficacy of policies or practices in achieving a reduction in environmental stressors have been removed.

2. Reviewer comment: How was macroinvertebrate sampling effort (to include enumeration effort) standardized across samples? Were adjustments made during statistical analysis where appropriate to account for any influence of variable sample effort? The macroinvertebrate enumeration protocol described in the manuscript appears to allow for a variable level of effort in enumeration (I trust that sample collection effort per Spanish protocol is standardized among sites; if not, that is of concern as well and should be addressed). As written, the manuscript relates an unclear and potentially inconsistent sub-sampling approach that could yield samples with vastly different number of individuals (level of effort) per sample. As written, the text suggests that some samples may have been sub-sampled, some may not have been, and the criteria are not clearly presented that were used for deciding when to subsample and when to stop counting after 300 individuals. As greater effort would naturally yield greater taxa richness (with implications for statistical analysis and interpretation), it is important to ensure that any observed increase in richness is not an artifact of the sampling procedure.

The macroinvertebrate sampling and identification protocol has been described in the new version of the manuscript according to the reviewer’s comments. This quantitative protocol considers the presence of 5 different habitat types: hard substrate, fine sediments, plant debris, submerged bank vegetation and submerged macrophytes. Then, 20 surber samples (0.1 m2 surface) are taken, proportionally distributed among each type of habitat according to the percentage of streambed surface occupied by each of them. For counting macroinvertebrates, sub-sampling was used to estimate the taxa abundances once at least 300 individuals per sample were counted, although sample exploration continued to check for new taxa.

3. Reviewer comment: Results section could be improved to better convey synthesized findings, as opposed to summary of measurements (which are in the Tables). By that I mean, tell the reader what you found; what should the reader take away after reading Results? To me, this section would be more compelling if it focused on conveying the findings that certain abiotic factors differed between biologically-different groups 1 and 2.

Additionally, the section could benefit from a bit more context; frame values relative to what is expected in streams of the region with minimal impact so the reader understands whether those values are of concern.

As well, or at the least, I suggest rearranging all presentation of data in tables, and possibly in text, to be grouped by the site groupings that were developed through cluster analysis (as opposed to the alphabetical arrangement at present). As those groupings are used for Fig 3 and Fig 4, seeing the abiotic data in tables grouped accordingly should facilitate easier contemplation of associations between abiotic and biotic data. For example, it is notable that, with a few exceptions, sites in Group 1 have lower specific conductance than sites in Group 2. Such an adjustment could also work well to more clearly convey differences between groups as presented in Results L211-261 and make the entire section more compelling. Given that you have established that biological differences exist between groups (Fig 2, Fig 3), presenting the abiotic information accordingly is more ecologically meaningful than site abbreviations that mean nothing to the unacquainted reader.

We have synthesized the results section and focused on findings that were relevant for the objectives and discussion of the paper. Standards for the environmental variables and biotic indices included in the Spanish legislation (RD 817/2015), based on WFD, are shown in Table 1, as well as the ranges of the IHF and QBR indices. We have also rearranged data presentation (e.g. grouping sites according to groups obtained in cluster analysis and focusing on differences between established groups).

4. Reviewer comment: Regarding averaging physicochemical parameters (L191-193, L225, and Tables 1 and S3), I generally agree that an annual mean derived from quarterly sampling can give a better long-term or overall representation of conditions to which organisms are exposed as compared to just one season of data. However, such averaging can mask seasonally-variable parameters that may have different effects at different seasons.

Summer, for example, is a particularly stressful time of year as it combines heat stress (summer maximum temperature) with low dissolved oxygen saturation potential (resulting from said heat). Similarly, nutrient levels are potentially of greater influence during the growing season (or leaf-off season) when sunlight and/or temperature are at levels sufficient to induce rapid excessive algal growth. Further, presenting the standard error of the mean in Table 1 is of marginal utility in describing values that vary widely across seasons (as noted with nitrate on L236-237). I’d prefer to see mean and range, the latter of which is helpful for understanding if extreme values (e.g., low summer oxygen, high summer temperature) may be influencing biota.

According to reviewer´s suggestion, we show the mean and the range (i.e. minimum and maximum) of the physicochemical variables.

MINOR COMMENTS

1. Reviewer comment: On what basis has sedimentation been parsed from other factors in this study as a main driver of degradation? The largest proportion of variation was noted as the aggregate IHF-total, and no sediment-specific parameters are included in the list of other significant explanatory factors (L270-279). Presumably sediment components of the IHF were included in DISTLM just as was done with QBR-natur, yet those IHF components were not prominent factors differentiating groups. I don’t think the data and analyses here support such a narrow conclusion as written.

We considered sedimentation as the main driver of degradation in sites affected by intensive agriculture because of two main issues:

a. The IHF index explained the largest proportion of variation among sites according to the DISTLM analysis. The first component of this index (IHF-1) refers to hard substrate embeddedness in fine sediment, although it resulted significantly correlated with the IHF total index (i.e. Spearman correlation coefficient higher than 0.9) and it was consequently removed from the DISTLM analysis. Anyway, as it can be seen in table S4, every site of group 2 has a score of 0 in this IHF component, which means that more than 60% of hard substrate was embedded in fine sediment; whereas every site of group 1 showed lower embeddedness and, consequently a higher score of this component.

b. Moreover, as we now state in the discussion, several traits significantly associated with the sites more impacted by agriculture seemed to be related to sediment increase. Accordingly, Murphy et al. (2017) confirmed the potential of macroinvertebrate biological traits as indicators of fine sediment impacts on rivers.

Murphy JF, Jones JI, Arnold A, Duerdoth CP, Pretty JL, Naden PS, et al. Can macroinvertebrate biological traits indicate fine-grained sediment conditions in streams? River Res Appl. 2017;33(10):1606–17

In any case, we have tried to avoid overstatements along the discussion section in the new version of the manuscript.

2. Reviewer comment: Language in several parts of the discussion is too conclusory regarding causation for an observational study. Please restate to reflect the correlative nature of the study and analyses.

Following the reviewer’s suggestions, we have reworded several sentences of the discussion section to make them less conclusory and acknowledge the limitations of our study.

---

## [Decision Letter · Decision Letter 1]

17 Oct 2019

PONE-D-19-15451R1

Agricultural impacts on streams near Nitrate Vulnerable Zones: a case study in the Ebro basin, northern Spain

PLOS ONE

Dear Dr. Cañedo-Argüelles Iglesias,

Thank you for submitting your manuscript to PLOS ONE. After careful consideration, we feel that it has merit but does not fully meet PLOS ONE’s publication criteria as it currently stands. Therefore, we invite you to submit a revised version of the manuscript that addresses the points raised during the review process.

Overall, all comments have been carefully addressed and the manuscript is now improved and more specific. Please, consider the minor comments below. 

Some additional minor comments:

Table 1: Please, use the same units for all concentrations either express O2 in ppm or the other nutrients in mg/L. 

Throughout the text replace NO3 by NO_3_ ^−^

We would appreciate receiving your revised manuscript by Dec 01 2019 11:59PM. To enhance the reproducibility of your results, we recommend that if applicable you deposit your laboratory protocols in protocols.io, where a protocol can be assigned its own identifier (DOI) such that it can be cited independently in the future. For instructions see: http://journals.plos.org/plosone/s/submission-guidelines#loc-laboratory-protocols

We look forward to receiving your revised manuscript.

Kind regards,

Clara Mendoza-Lera

Academic Editor

PLOS ONE

Reviewers' comments:

Reviewer's Responses to Questions

**Comments to the Author**

1. If the authors have adequately addressed your comments raised in a previous round of review and you feel that this manuscript is now acceptable for publication, you may indicate that here to bypass the “Comments to the Author” section, enter your conflict of interest statement in the “Confidential to Editor” section, and submit your "Accept" recommendation.

Reviewer #1: All comments have been addressed

Reviewer #3: (No Response)

Reviewer #5: All comments have been addressed

2. Is the manuscript technically sound, and do the data support the conclusions?

Reviewer #1: Yes

Reviewer #3: Yes

Reviewer #5: Yes

3. Has the statistical analysis been performed appropriately and rigorously? 

Reviewer #1: Yes

Reviewer #3: Yes

Reviewer #5: Yes

4. Have the authors made all data underlying the findings in their manuscript fully available?

Reviewer #1: Yes

Reviewer #3: Yes

Reviewer #5: Yes

5. Is the manuscript presented in an intelligible fashion and written in standard English?

Reviewer #1: Yes

Reviewer #3: Yes

Reviewer #5: Yes

6. Review Comments to the Author

Reviewer #1: The authors addressed all the comments from the previous revision round. I have only a few further comments:

Lines 245-246: What variables were removed because of the correlation with other variables?

Lines 314-315: “Group 1 was characterized by taxa with short reproductive cycles” [] “Group 2 was characterized by taxa with longer reproductive cycles”. This is wrong, if the studied trait represents the number of reproductive cycles per year, as for line 203. The asymmetric beanplots show that Group 1 has higher proportion of the modality “less than 1” and “1” compared to Group 2; and Group 2 has higher proportion of the modality “more than 1” compared to Group1. I´d recommend to describe them as mono- or semivoltine taxa (group 1) and polyvoltine taxa (group 2), rather than taxa with long/short reproductive cycle.

Check figure numbering. The text refers to the asymmetric beanplots as Figure 3, but it was uploaded as Figure 4. The dbRDA plot was upleaded as Figure 3, instead of Figure 4.

Remove “all of them” at line 124.

Reviewer #3: The revised manuscript reads much better and I am satisfied that the authors have addressed their key prediction about the influence of different stressors associated with agricultural activities in their study region. I only have some minor corrections to suggest:

1. Abstract, L36: “produced” = “caused”

2. Abstract, L45: “good” = “useful” (since “good” is a WFD status class, it could get confusing).

3. Abstract, L47: Consider “The streams affected by a greater percentage of agricultural land cover in the surrounding catchment had significantly higher nitrate concentrations than the remaining sites”.

4. Abstract, L51: Consider adding quotation marks around “good agricultural practices”

5. Intro, L87: “Various studies suggest that 10 ppm…”

6. Intro, L104: “grazers” – This FFG probably reflects how different taxa are assigned (i.e., there is a separate category for “scrapers”), but an increase in “grazers” is at odds with “reducing substrate availability for grazing mayflies” as per smothering algae effects (L96). I´m not certain much can be done about this apparent contradiction, although the authors should be citing Rabeni et al. (2005) here and in the Discussion – and those researchers found that scrapers (presumably including grazers) decreased with increasing sedimentation. Note also that a decrease in shredders as per the expectation at L114-119 is at odds with what Rabeni et al. 2005 found; although Burdon et al. 2018 found that grazers (including scrapers) and shredders decreased across a landscape disturbance gradient whilst collectors become proportionally more abundant.

Rabení, C.F., Doisy, K.E. & Zweig, L.D. 2005. Aquatic Science 67: 395-402; https://doi.org/10.1007/s00027-005-0793-2

Burdon, F. J., McIntosh, A.R., Harding J.S. (2018). Mechanisms of trophic niche compression: evidence from landscape disturbance bioRxiv 329623; doi: https://doi.org/10.1101/329623 (Accepted for publication in the Journal of Animal Ecology).

7. Methods, L150: “studied” = “sampled”.

8. Methods, L168: “associated to” = “associated with”

9. Methods, L179: Capital S in “Surber”

10. Methods, L236: “distance-based”

11. Methods, L238: “abundances”

12. Results, L275: “concentrations”

13. Results, L308-311, Table 2: Check “Leptophlebiidae” – I guess the # doesn´t belong here

14. Results, L327: define “agricultural” land here – i.e., arable cropping, etc. “Pasture” is mentioned at L332 which is also an agricultural land use.

15. Conclusion: L441: “good” = “useful” (since good is also a WFD status category)

Reviewer #5: (No Response)

7. PLOS authors have the option to publish the peer review history of their article (what does this mean?). If published, this will include your full peer review and any attached files.

Reviewer #1: No

Reviewer #3: No

Reviewer #5: No

---

## [Author Response · Author response to Decision Letter 1]

21 Oct 2019

. Reviewer comment: Throughout the text replace NO3 by NO3−

Response: We have replaced NO3 by NO3− at manuscript, and accordingly, at Supporting Information (as the other ions) and at figure 4.

REVIEWER 1

1. Reviewer comment: What variables were removed because of the correlation with other variables?

Response: Concentration of SO42-, Ca2+ and Mg2+, Components 1, 2 and 7 of IHF index, QBR-total and Upstream siliceous rocks (%)

2. Reviewer comment: “Group 1 was characterized by taxa with short reproductive cycles”, “Group 2 was characterized by taxa with longer reproductive cycles”. This is wrong, if the studied trait represents the number of reproductive cycles per year, as for line 203. The asymmetric beanplots show that Group 1 has higher proportion of the modality “less than 1” and “1” compared to Group 2; and Group 2 has higher proportion of the modality “more than 1” compared to Group1. I´d recommend to describe them as mono- or semivoltine taxa (group 1) and polyvoltine taxa (group 2), rather than taxa with long/short reproductive cycle. 

Response: We are sorry for this mistake. According to reviewer recommendations, we have described taxa in the test as mono-, semi- or polyvoltine taxa, and we have accordingly added these terms in the Supporting Information.

3. Reviewer comment: Check figure numbering. The text refers to the asymmetric beanplots as Figure 3, but it was uploaded as Figure 4. The dbRDA plot was upleaded as Figure 3, instead of Figure 4.

Response: We uploaded now asymmetric beanplots as figure 3 and dbRDA plot as figure 4, as it is correct.

REVIEWER 3

1. Reviewer comment: “grazers” – This FFG probably reflects how different taxa are assigned (i.e., there is a separate category for “scrapers”), but an increase in “grazers” is at odds with “reducing substrate availability for grazing mayflies” as per smothering algae effects (L96). I´m not certain much can be done about this apparent contradiction, although the authors should be citing Rabeni et al. (2005) here and in the Discussion – and those researchers found that scrapers (presumably including grazers) decreased with increasing sedimentation. Note also that a decrease in shredders as per the expectation at L114-119 is at odds with what Rabeni et al. 2005 found; although Burdon et al. 2018 found that grazers (including scrapers) and shredders decreased across a landscape disturbance gradient whilst collectors become proportionally more abundant. 

Response: We have cited now Rabeni et al. (2005) and Burdon et al. (2018) in the introduction and in the discussion. Both studies are in line with our findings of scrapers decrease under sediment increase.

Regarding shredders, we are not sure why Rabení and Burdon find opposite patterns, but we have suggested the following: “The abundance of shredders was also reduced under agricultural intensification. As suggested by Burdon et al. [27], this is most likely due to degraded riparian habitats (Fig 5), since shredders do not seem to be directly affected by sediment deposition [33]”.

2. Reviewer comment: define “agricultural” land here – i.e., arable cropping, etc. “Pasture” is mentioned at L332 which is also an agricultural land use

Response: we have detailed that agriculture land refers to cultivated land when it is firstly cited in the manuscript.

3. Reviewer comment: Table 2: Check “Leptophlebiidae” – I guess the # doesn´t belong here. 

Response: # symbol is correctly included here, since this family was significantly associated with group 1 according to IndVal analysis, but Habroleptoides sp. and Habrophlebia sp. were not significantly associated to any group according to this analysis, probably related to their lower densities.

---

## [Editor Report · Decision Letter 2]

23 Oct 2019

Agricultural impacts on streams near Nitrate Vulnerable Zones: a case study in the Ebro basin, northern Spain

PONE-D-19-15451R2

Dear Dr. Cañedo-Argüelles Iglesias,

We are pleased to inform you that your manuscript has been judged scientifically suitable for publication and will be formally accepted for publication once it complies with all outstanding technical requirements.

With kind regards,

Clara Mendoza-Lera

Academic Editor

PLOS ONE
---

## [Editor Report · Acceptance letter]

1 Nov 2019

PONE-D-19-15451R2 

Agricultural impacts on streams near Nitrate Vulnerable Zones: a case study in the Ebro basin, northern Spain 

Dear Dr. Cañedo-Argüelles Iglesias:

I am pleased to inform you that your manuscript has been deemed suitable for publication in PLOS ONE. Congratulations! Your manuscript is now with our production department. 

With kind regards,

on behalf of

Dr. Clara Mendoza-Lera 

Academic Editor

PLOS ONE